# Generalization Bounds of Stochastic Gradient Descent for Wide and Deep Neural Networks

**Yuan Cao**
Department of Computer Science
University of California, Los Angeles
CA 90095, USA
yuancao@cs.ucla.edu

**Quanquan Gu**
Department of Computer Science
University of California, Los Angeles
CA 90095, USA
qgu@cs.ucla.edu

## Abstract

We study the training and generalization of deep neural networks (DNNs) in the over-parameterized regime, where the network width (i.e., number of hidden nodes per layer) is much larger than the number of training data points. We show that, the expected 0-1 loss of a wide enough ReLU network trained with stochastic gradient descent (SGD) and random initialization can be bounded by the training loss of a random feature model induced by the network gradient at initialization, which we call a *neural tangent random feature* (NTRF) model. For data distributions that can be classified by NTRF model with sufficiently small error, our result yields a generalization error bound in the order of $\widetilde{\mathcal{O}}(n^{-1/2})$ that is independent of the network width. Our result is more general and sharper than many existing generalization error bounds for over-parameterized neural networks. In addition, we establish a strong connection between our generalization error bound and the neural tangent kernel (NTK) proposed in recent work.

## 1 Introduction

Deep learning has achieved great success in a wide range of applications including image processing [20], natural language processing [17] and reinforcement learning [34]. Most of the deep neural networks used in practice are highly over-parameterized, such that the number of parameters is much larger than the number of training data. One of the mysteries in deep learning is that, even in an over-parameterized regime, neural networks trained with stochastic gradient descent can still give small test error and do not overfit. In fact, a famous empirical study by Zhang et al. [38] shows the following phenomena:

- Even if one replaces the real labels of a training data set with purely random labels, an over-parameterized neural network can still fit the training data perfectly. However since the labels are independent of the input, the resulting neural network does not generalize to the test dataset.

- If the same over-parameterized network is trained with real labels, it not only achieves small training loss, but also generalizes well to the test dataset.

While a series of recent work has theoretically shown that a sufficiently over-parameterized (i.e., sufficiently wide) neural network can fit random labels [12, 2, 11, 39], the reason why it can generalize well when trained with real labels is less understood. Existing generalization bounds for deep neural networks [29, 6, 27, 15, 13, 5, 24, 35, 28] based on uniform convergence usually cannot provide non-vacuous bounds [21, 13] in the over-parameterized regime. In fact, the empirical observation by Zhang et al. [38] indicates that in order to understand deep learning, it is important to distinguish the true data labels from random labels when studying generalization. In other words, it is essential to quantify the "classifiability" of the underlying data distribution, i.e., how difficult it can be classified.

Certain effort has been made to take the "classifiability" of the data distribution into account for generalization analysis of neural networks. Brutzkus et al. [7] showed that stochastic gradient descent (SGD) can learn an over-parameterized two-layer neural network with good generalization for linearly separable data. Li and Liang [25] proved that, if the data satisfy certain structural assumption, SGD can learn an over-parameterized two-layer network with fixed second layer weights and achieve a small generalization error. Allen-Zhu et al. [1] studied the generalization performance of SGD and its variants for learning two-layer and three-layer networks, and used the risk of smaller two-layer or three-layer networks with smooth activation functions to characterize the classifiability of the data distribution. There is another line of studies on the algorithm-dependent generalization bounds of neural networks in the over-parameterized regime [10, 4, 8, 37, 14], which quantifies the classifiability of the data with a reference function class defined by random features [31, 32] or kernels[1]. Specifically, Daniely [10] showed that a neural network of large enough size is competitive with the best function in the conjugate kernel class of the network. Arora et al. [4] gave a generalization error bound for two-layer ReLU networks with fixed second layer weights based on a ReLU kernel function. Cao and Gu [8] showed that deep ReLU networks trained with gradient descent can achieve small generalization error if the data can be separated by certain random feature model [32] with a margin. Yehudai and Shamir [37] used the expected loss of a similar random feature model to quantify the generalization error of two-layer neural networks with smooth activation functions. A similar generalization error bound was also given by E et al. [14], where the authors studied the optimization and generalization of two-layer networks trained with gradient descent. However, all the aforementioned results are still far from satisfactory: they are either limited to two-layer networks, or restricted to very simple and special reference function classes.

In this paper, we aim at providing a sharper and generic analysis on the generalization of deep ReLU networks trained by SGD. In detail, we base our analysis upon the key observations that near random initialization, the neural network function is almost a linear function of its parameters and the loss function is locally almost convex. This enables us to prove a cumulative loss bound of SGD, which further leads to a generalization bound by online-to-batch conversion [9]. The main contributions of our work are summarized as follows:

- We give a bound on the expected 0-1 error of deep ReLU networks trained by SGD with random initialization. Our result relates the generalization bound of an over-parameterized ReLU network with a random feature model defined by the network gradients, which we call *neural tangent random feature* (NTRF) model. It also suggests an algorithm-dependent generalization error bound of order $\widetilde{\mathcal{O}}(n^{-1/2})$, which is independent of network width, if the data can be classified by the NTRF model with small enough error.

- Our analysis is general enough to cover recent generalization error bounds for neural networks with random feature based reference function classes, and provides better bounds. Our expected 0-1 error bound directly covers the result by Cao and Gu [8], and gives a tighter sample complexity when reduced to their setting, i.e., $\widetilde{\mathcal{O}}(1/\epsilon^2)$ versus $\widetilde{\mathcal{O}}(1/\epsilon^4)$ where $\epsilon$ is the target generalization error. Compared with recent results by Yehudai and Shamir [37], E et al. [14] who only studied two-layer networks, our bound not only works for deep networks, but also uses a larger reference function class when reduced to the two-layer setting, and therefore is sharper.

- Our result has a direct connection to the neural tangent kernel studied in Jacot et al. [18]. When interpreted in the language of kernel method, our result gives a generalization bound in the form of $\widetilde{\mathcal{O}}(L \cdot \sqrt{\mathbf{y}^\top (\mathbf{\Theta}^{(L)})^{-1} \mathbf{y}/n})$, where $\mathbf{y}$ is the training label vector, and $\mathbf{\Theta}^{(L)}$ is the neural tangent kernel matrix defined on the training input data. This form of generalization bound is similar to, but more general and tighter than the bound given by Arora et al. [4].

**Notation** We use lower case, lower case bold face, and upper case bold face letters to denote scalars, vectors and matrices respectively. For a vector $\mathbf{v} = (v_1, \ldots, v_d)^T \in \mathbb{R}^d$ and a number $1 \leqslant p < \infty$, let $\|\mathbf{v}\|_p = (\sum_{i=1}^d |v_i|^p)^{1/p}$. We also define $\|\mathbf{v}\|_\infty = \max_i |v_i|$. For a matrix $\mathbf{A} = (A_{i,j})_{m \times n}$, we use $\|\mathbf{A}\|_0$ to denote the number of non-zero entries of $\mathbf{A}$, and denote $\|\mathbf{A}\|_F = (\sum_{i,j=1}^d A_{i,j}^2)^{1/2}$ and $\|\mathbf{A}\|_p = \max_{\|\mathbf{v}\|_p=1} \|\mathbf{A}\mathbf{v}\|_p$ for $p \geqslant 1$. For two matrices $\mathbf{A}, \mathbf{B} \in \mathbb{R}^{m \times n}$, we define $\langle \mathbf{A}, \mathbf{B} \rangle = \mathrm{Tr}(\mathbf{A}^\top \mathbf{B})$. We denote by $\mathbf{A} \succeq \mathbf{B}$ if $\mathbf{A} - \mathbf{B}$ is positive semidefinite. In addition, we define the

asymptotic notations $\mathcal{O}(\cdot)$, $\widetilde{\mathcal{O}}(\cdot)$, $\Omega(\cdot)$ and $\widetilde{\Omega}(\cdot)$ as follows. Suppose that $a_n$ and $b_n$ be two sequences. We write $a_n = \mathcal{O}(b_n)$ if $\limsup_{n\to\infty} |a_n/b_n| < \infty$, and $a_n = \Omega(b_n)$ if $\liminf_{n\to\infty} |a_n/b_n| > 0$. We use $\widetilde{\mathcal{O}}(\cdot)$ and $\widetilde{\Omega}(\cdot)$ to hide the logarithmic factors in $\mathcal{O}(\cdot)$ and $\Omega(\cdot)$.

## 2 Problem Setup

In this section we introduce the basic problem setup. Following the same standard setup implemented in the line of recent work [2, 11, 39, 8], we consider fully connected neural networks with width $m$, depth $L$ and input dimension $d$. Such a network is defined by its weight matrices at each layer: for $L \geqslant 2$, let $\mathbf{W}_1 \in \mathbb{R}^{m \times d}$, $\mathbf{W}_l \in \mathbb{R}^{m \times m}$, $l = 2, \ldots, L-1$ and $\mathbf{W}_L \in \mathbb{R}^{1 \times m}$ be the weight matrices of the network. Then the neural network with input $\mathbf{x} \in \mathbb{R}^d$ is defined as

$$f_{\mathbf{W}}(\mathbf{x}) = \sqrt{m} \cdot \mathbf{W}_L \sigma(\mathbf{W}_{L-1} \sigma(\mathbf{W}_{L-2} \cdots \sigma(\mathbf{W}_1 \mathbf{x}) \cdots)), \tag{2.1}$$

where $\sigma(\cdot)$ is the entry-wise activation function. In this paper, we only consider the ReLU activation function $\sigma(z) = \max\{0, z\}$, which is the most commonly used activation function in applications. It is also arguably one of the most difficult activation functions to analyze, due to its non-smoothess. We remark that our result can be generalized to many other Lipschitz continuous and smooth activation functions. For simplicity, we follow Allen-Zhu et al. [2], Du et al. [11] and assume that the widths of each hidden layer are the same. Our result can be easily extended to the setting that the widths of each layer are not equal but in the same order, as discussed in Zou et al. [39], Cao and Gu [8].

When $L = 1$, the neural network reduces to a linear function, which has been well-studied. Therefore, for notational simplicity we focus on the case $L \geqslant 2$, where the parameter space is defined as

$$\mathcal{W} := \mathbb{R}^{m \times d} \times (\mathbb{R}^{m \times m})^{L-2} \times \mathbb{R}^{1 \times m}.$$

We also use $\mathbf{W} = (\mathbf{W}_1, \ldots, \mathbf{W}_L) \in \mathcal{W}$ to denote the collection of weight matrices for all layers. For $\mathbf{W}, \mathbf{W}' \in \mathcal{W}$, we define their inner product as $\langle \mathbf{W}, \mathbf{W}' \rangle := \sum_{l=1}^{L} \text{Tr}(\mathbf{W}_l^\top \mathbf{W}_l')$.

The goal of neural network learning is to minimize the expected risk, i.e.,

$$\min_{\mathbf{W}} L_{\mathcal{D}}(\mathbf{W}) := \mathbb{E}_{(\mathbf{x},y)\sim\mathcal{D}} L_{(\mathbf{x},y)}(\mathbf{W}), \tag{2.2}$$

where $L_{(\mathbf{x},y)}(\mathbf{W}) = \ell[y \cdot f_{\mathbf{W}}(\mathbf{x})]$ is the loss defined on any example $(\mathbf{x}, y)$, and $\ell(z)$ is the loss function. Without loss of generality, we consider the cross-entropy loss in this paper, which is defined as $\ell(z) = \log[1 + \exp(-z)]$. We would like to emphasize that our results also hold for most convex and Lipschitz continuous loss functions such as hinge loss. We now introduce stochastic gradient descent based training algorithm for minimizing the expected risk in (2.2). The detailed algorithm is given in Algorithm 1.

---

**Algorithm 1** SGD for DNNs starting at Gaussian initialization

---

    **Input:** Number of iterations $n$, step size $\eta$.
    Generate each entry of $\mathbf{W}_l^{(0)}$ independently from $N(0, 2/m)$, $l \in [L-1]$.
    Generate each entry of $\mathbf{W}_L^{(0)}$ independently from $N(0, 1/m)$.
    **for** $i = 1, 2, \ldots, n$ **do**
        Draw $(\mathbf{x}_i, y_i)$ from $\mathcal{D}$.
        Update $\mathbf{W}^{(i)} = \mathbf{W}^{(i-1)} - \eta \cdot \nabla_{\mathbf{W}} L_{(\mathbf{x}_i, y_i)}(\mathbf{W}^{(i)})$.
    **end for**
    **Output:** Randomly choose $\widehat{\mathbf{W}}$ uniformly from $\{\mathbf{W}^{(0)}, \ldots, \mathbf{W}^{(n-1)}\}$.

---

The initialization scheme for $\mathbf{W}^{(0)}$ given in Algorithm 1 generates each entry of the weight matrices from a zero-mean independent Gaussian distribution, whose variance is determined by the rule that the expected length of the output vector in each hidden layer is equal to the length of the input. This initialization method is also known as He initialization [16]. Here the last layer parameter is initialized with variance $1/m$ instead of $2/m$ since the last layer is not associated with the ReLU activation function.

# 3 Main Results

In this section we present the main results of this paper. In Section 3.1 we give an expected 0-1 error bound against a neural tangent random feature reference function class. In Section 3.2, we discuss the connection between our result and the neural tangent kernel proposed in Jacot et al. [18].

## 3.1 An Expected 0-1 Error Bound

In this section we give a bound on the expected 0-1 error $L_{\mathcal{D}}^{0-1}(\mathbf{W}) := \mathbb{E}_{(\mathbf{x},y) \sim \mathcal{D}}[\mathbb{1}\{y \cdot f_{\mathbf{W}}(\mathbf{x}) < 0\}]$ obtained by Algorithm 1. Our result is based on the following assumption.

**Assumption 3.1.** The data inputs are normalized: $\|\mathbf{x}\|_2 = 1$ for all $(\mathbf{x}, y) \in \text{supp}(\mathcal{D})$.

Assumption 3.1 is a standard assumption made in almost all previous work on optimization and generalization of over-parameterized neural networks [12, 2, 11, 39, 30, 14]. As is mentioned in Cao and Gu [8], this assumption can be relaxed to $c_1 \leqslant \|\mathbf{x}\|_2 \leqslant c_2$ for all $(\mathbf{x}, y) \in \text{supp}(\mathcal{D})$, where $c_2 > c_1 > 0$ are absolute constants.

For any $\mathbf{W} \in \mathcal{W}$, we define its $\omega$-neighborhood as

$$\mathcal{B}(\mathbf{W}, \omega) := \{\mathbf{W}' \in \mathcal{W} : \|\mathbf{W}'_l - \mathbf{W}_l\|_F \leqslant \omega, l \in [L]\}.$$

Below we introduce the neural tangent random feature function class, which serves as a reference function class to measure the "classifiability" of the data, i.e., how easy it can be classified.

**Definition 3.2** (Neural Tangent Random Feature). Let $\mathbf{W}^{(0)}$ be generated via the initialization scheme in Algorithm 1. The neural tangent random feature (NTRF) function class is defined as

$$\mathcal{F}(\mathbf{W}^{(0)}, R) = \{f(\cdot) = f_{\mathbf{W}^{(0)}}(\cdot) + \langle \nabla_{\mathbf{W}} f_{\mathbf{W}^{(0)}}(\cdot), \mathbf{W} \rangle : \mathbf{W} \in \mathcal{B}(\mathbf{0}, R \cdot m^{-1/2})\},$$

where $R > 0$ measures the size of the function class, and $m$ is the width of the neural network.

The name "neural tangent random feature" is inspired by the neural tangent kernel proposed by Jacot et al. [18], because the random features are the gradients of the neural network with random weights. Connections between the neural tangent random features and the neural tangent kernel will be discussed in Section 3.2.

We are ready to present our main result on the expected 0-1 error bound of Algorithm 1.

**Theorem 3.3.** For any $\delta \in (0, e^{-1}]$ and $R > 0$, there exists

$$m^*(\delta, R, L, n) = \tilde{\mathcal{O}}\big(\text{poly}(R, L)\big) \cdot n^7 \cdot \log(1/\delta)$$

such that if $m \geqslant m^*(\delta, R, L, n)$, then with probability at least $1 - \delta$ over the randomness of $\mathbf{W}^{(0)}$, the output of Algorithm 1 with step size $\eta = \kappa \cdot R/(m\sqrt{n})$ for some small enough absolute constant $\kappa$ satisfies

$$\mathbb{E}\big[L_{\mathcal{D}}^{0-1}(\widehat{\mathbf{W}})\big] \leqslant \inf_{f \in \mathcal{F}(\mathbf{W}^{(0)}, R)} \left\{ \frac{4}{n} \sum_{i=1}^{n} \ell[y_i \cdot f(\mathbf{x}_i)] \right\} + \mathcal{O}\left[\frac{LR}{\sqrt{n}} + \sqrt{\frac{\log(1/\delta)}{n}}\right], \qquad (3.1)$$

where the expectation is taken over the uniform draw of $\widehat{\mathbf{W}}$ from $\{\mathbf{W}^{(0)}, \ldots, \mathbf{W}^{(n-1)}\}$.

The expected 0-1 error bound given by Theorem 3.3 consists of two terms: The first term in (3.1) relates the expected 0-1 error achieved by Algorithm 1 with a reference function class–the NTRF function class in Definition 3.2. The second term in (3.1) is a standard large-deviation error term. As long as $R = \tilde{\mathcal{O}}(1)$, this term matches the standard $\tilde{\mathcal{O}}(n^{-1/2})$ rate in PAC learning bounds [33].

**Remark 3.4.** The parameter $R$ in Theorem 3.3 is from the NTRF class and introduces a trade-off in the bound: when $R$ is small, the corresponding NTRF class $\mathcal{F}(\mathbf{W}^{(0)}, R)$ is small, making the first term in (3.1) large, and the second term in (3.1) is small. When $R$ is large, the corresponding function class $\mathcal{F}(\mathbf{W}^{(0)}, R)$ is large, so the first term in (3.1) is small, and the second term will be large. In particular, if we set $R = \tilde{\mathcal{O}}(1)$, the second term in (3.1) will be $\tilde{\mathcal{O}}(n^{-1/2})$. In this case, the "classifiability" of the underlying data distribution $\mathcal{D}$ is determined by how well its i.i.d. samples can be classified by $\mathcal{F}(\mathbf{W}^{(0)}, \tilde{\mathcal{O}}(1))$. In other words, Theorem 3.3 suggests that if the data can be classified by a function in the NTRF function class $\mathcal{F}(\mathbf{W}^{(0)}, \tilde{\mathcal{O}}(1))$ with a small training error, the over-parameterized ReLU network learnt by Algorithm 1 will have a small generalization error.

**Remark 3.5.** The expected 0-1 error bound given by Theorem 3.3 is in a very general form. It directly covers the result given by Cao and Gu [8]. In Appendix A.1, we show that under the same assumptions made in Cao and Gu [8], to achieve $\epsilon$ expected 0-1 error, our result requires a sample complexity of order $\widetilde{\mathcal{O}}(\epsilon^{-2})$, which outperforms the result in Cao and Gu [8] by a factor of $\epsilon^{-2}$.

**Remark 3.6.** Our generalization bound can also be compared with two recent results [37, 14] for two-layer neural networks. When $L = 2$, the NTRF function class $\mathcal{F}(\mathbf{W}^{(0)}, \widetilde{\mathcal{O}}(1))$ can be written as

$$\big\{ f_{\mathbf{W}^{(0)}}(\cdot) + \langle \nabla_{\mathbf{W}_1} f_{\mathbf{W}^{(0)}}(\cdot), \mathbf{W}_1 \rangle + \langle \nabla_{\mathbf{W}_2} f_{\mathbf{W}^{(0)}}(\cdot), \mathbf{W}_2 \rangle : \|\mathbf{W}_1\|_F, \|\mathbf{W}_2\|_F \leqslant \widetilde{\mathcal{O}}(m^{-1/2}) \big\}.$$

In contrast, the reference function classes studied by Yehudai and Shamir [37] and E et al. [14] are contained in the following random feature class:

$$\mathcal{F} = \big\{ f_{\mathbf{W}^{(0)}}(\cdot) + \langle \nabla_{\mathbf{W}_2} f_{\mathbf{W}^{(0)}}(\cdot), \mathbf{W}_2 \rangle : \|\mathbf{W}_2\|_F \leqslant \widetilde{\mathcal{O}}(m^{-1/2}) \big\},$$

where $\mathbf{W}^{(0)} = (\mathbf{W}_1^{(0)}, \mathbf{W}_2^{(0)}) \in \mathbb{R}^{m \times d} \times \mathbb{R}^{1 \times m}$ are the random weights generated by the initialization schemes in Yehudai and Shamir [37], E et al. [14][2]. Evidently, our NTRF function class $\mathcal{F}(\mathbf{W}^{(0)}, \widetilde{\mathcal{O}}(1))$ is richer than $\mathcal{F}$–it also contains the features corresponding to the first layer gradient of the network at random initialization, i.e., $\nabla_{\mathbf{W}_1} f_{\mathbf{W}^{(0)}}(\cdot)$. As a result, our generalization bound is sharper than those in Yehudai and Shamir [37], E et al. [14] in the sense that we can show that neural networks trained with SGD can compete with the best function in a larger reference function class.

As previously mentioned, the result of Theorem 3.3 can be easily extended to the setting where the widths of different layers are different. We should expect that the result remains almost the same, except that we assume the widths of hidden layers are all larger than or equal to $m^*(\delta, R, L, n)$. We would also like to point out that although this paper considers the cross-entropy loss, the proof of Theorem 3.3 offers a general framework based on the fact that near initialization, the neural network function is almost linear in terms of its weights. We believe that this proof framework can potentially be applied to most practically useful loss functions: whenever $\ell(\cdot)$ is convex/Lipschitz continuous/smooth, near initialization, $L_i(\mathbf{W})$ is also almost convex/Lipschitz continuous/smooth in $\mathbf{W}$ for all $i \in [n]$, and therefore standard online optimization analysis can be invoked with online-to-batch conversion to provide a generalization bound. We refer to Section 4 for more details.

### 3.2 Connection to Neural Tangent Kernel

Besides quantifying the classifiability of the data with the NTRF function class $\mathcal{F}(\mathbf{W}^{(0)}, \widetilde{\mathcal{O}}(1))$, an alternative way to apply Theorem 3.3 is to check how large the parameter $R$ needs to be in order to make the first term in (3.1) small enough (e.g., smaller than $n^{-1/2}$). In this subsection, we show that this type of analysis connects Theorem 3.3 to the neural tangent kernel proposed in Jacot et al. [18] and later studied by Yang [36], Lee et al. [23], Arora et al. [3]. Specifically, we provide an expected 0-1 error bound in terms of the neural tangent kernel matrix defined over the training data. We first define the neural tangent kernel matrix for the neural network function in (2.1).

**Definition 3.7** (Neural Tangent Kernel Matrix). For any $i, j \in [n]$, define

$$\widetilde{\mathbf{\Theta}}_{i,j}^{(1)} = \mathbf{\Sigma}_{i,j}^{(1)} = \langle \mathbf{x}_i, \mathbf{x}_j \rangle, \quad \mathbf{A}_{ij}^{(l)} = \begin{pmatrix} \mathbf{\Sigma}_{i,i}^{(l)} & \mathbf{\Sigma}_{i,j}^{(l)} \\ \mathbf{\Sigma}_{i,j}^{(l)} & \mathbf{\Sigma}_{j,j}^{(l)} \end{pmatrix},$$

$$\mathbf{\Sigma}_{i,j}^{(l+1)} = 2 \cdot \mathbb{E}_{(u,v) \sim N\left(\mathbf{0}, \mathbf{A}_{ij}^{(l)}\right)} [\sigma(u)\sigma(v)],$$

$$\widetilde{\mathbf{\Theta}}_{i,j}^{(l+1)} = \widetilde{\mathbf{\Theta}}_{i,j}^{(l)} \cdot 2 \cdot \mathbb{E}_{(u,v) \sim N\left(\mathbf{0}, \mathbf{A}_{ij}^{(l)}\right)} [\sigma'(u)\sigma'(v)] + \mathbf{\Sigma}_{i,j}^{(l+1)}.$$

Then we call $\mathbf{\Theta}^{(L)} = [(\widetilde{\mathbf{\Theta}}_{i,j}^{(L)} + \mathbf{\Sigma}_{i,j}^{(L)})/2]_{n \times n}$ the neural tangent kernel matrix of an $L$-layer ReLU network on training inputs $\mathbf{x}_1, \ldots, \mathbf{x}_n$.

Definition 3.7 is the same as the original definition in Jacot et al. [18] when restricting the kernel function on $\{\mathbf{x}_1, \ldots, \mathbf{x}_n\}$, except that there is an extra coefficient 2 in the second and third lines. This extra factor is due to the difference in initialization schemes–in our paper the entries of hidden layer

matrices are randomly generated with variance $2/m$, while in Jacot et al. [18] the variance of the random initialization is $1/m$. We remark that this extra factor 2 in Definition 3.7 will remove the exponential dependence on the network depth $L$ in the kernel matrix, which is appealing. In fact, it is easy to check that under our scaling, the diagonal entries of $\mathbf{\Sigma}^{(L)}$ are all 1's, and the diagonal entries of $\widetilde{\mathbf{\Theta}}^{(L)}$ are all $L$'s.

The following lemma is a summary of Theorem 1 and Proposition 2 in Jacot et al. [18], which ensures that $\mathbf{\Theta}^{(L)}$ is the infinite-width limit of the Gram matrix $(m^{-1}\langle \nabla_{\mathbf{W}} f_{\mathbf{W}^{(0)}}(\mathbf{x}_i), \nabla_{\mathbf{W}} f_{\mathbf{W}^{(0)}}(\mathbf{x}_j) \rangle)_{n \times n}$, and is positive-definite as long as no two training inputs are parallel.

**Lemma 3.8** (Jacot et al. [18]). For an $L$ layer ReLU network with parameter set $\mathbf{W}^{(0)}$ initialized in Algorithm 1, as the network width $m \to \infty$[3], it holds that

$$m^{-1}\langle \nabla_{\mathbf{W}} f_{\mathbf{W}^{(0)}}(\mathbf{x}_i), \nabla_{\mathbf{W}} f_{\mathbf{W}^{(0)}}(\mathbf{x}_j) \rangle \xrightarrow{\mathbb{P}} \mathbf{\Theta}^{(L)}_{i,j},$$

where the expectation is taken over the randomness of $\mathbf{W}^{(0)}$. Moreover, as long as each pair of inputs among $\mathbf{x}_1, \ldots, \mathbf{x}_n \in S^{d-1}$ are not parallel, $\mathbf{\Theta}^{(L)}$ is positive-definite.

**Remark 3.9.** Lemmas 3.8 clearly shows the difference between our neural tangent kernel matrix $\mathbf{\Theta}^{(L)}$ in Definition 3.7 and the Gram matrix $\mathbf{K}^{(L)}$ defined in Definition 5.1 in Du et al. [11]. For any $i, j \in [n]$, by Lemma 3.8 we have

$$\mathbf{\Theta}^{(L)}_{i,j} = \lim_{m \to \infty} m^{-1} \sum_{l=1}^{L} \langle \nabla_{\mathbf{W}_l} f_{\mathbf{W}^{(0)}}(\mathbf{x}_i), \nabla_{\mathbf{W}_l} f_{\mathbf{W}^{(0)}}(\mathbf{x}_j) \rangle.$$

In contrast, the corresponding entry in $\mathbf{K}^{(L)}$ is

$$\mathbf{K}^{(L)}_{i,j} = \lim_{m \to \infty} m^{-1} \langle \nabla_{\mathbf{W}_{L-1}} f_{\mathbf{W}^{(0)}}(\mathbf{x}_i), \nabla_{\mathbf{W}_{L-1}} f_{\mathbf{W}^{(0)}}(\mathbf{x}_j) \rangle.$$

It can be seen that our definition of kernel matrix takes all layers into consideration, while Du et al. [11] only considered the last hidden layer (i.e., second last layer). Moreover, it is clear that $\mathbf{\Theta}^{(L)} \succeq \mathbf{K}^{(L)}$. Since the smallest eigenvalue of the kernel matrix plays a key role in the analysis of optimization and generalization of over-parameterized neural networks [12, 11, 4], our neural tangent kernel matrix can potentially lead to better bounds than the Gram matrix studied in Du et al. [11].

**Corollary 3.10.** Let $\mathbf{y} = (y_1, \ldots, y_n)^\top$ and $\lambda_0 = \lambda_{\min}(\mathbf{\Theta}^{(L)})$. For any $\delta \in (0, e^{-1}]$, there exists $\widetilde{m}^*(\delta, L, n, \lambda_0)$ that only depends on $\delta, L, n$ and $\lambda_0$ such that if $m \geq \widetilde{m}^*(\delta, L, n, \lambda_0)$, then with probability at least $1 - \delta$ over the randomness of $\mathbf{W}^{(0)}$, the output of Algorithm 1 with step size $\eta = \kappa \cdot \inf_{\widetilde{y}_i y_i \geq 1} \sqrt{\widetilde{\mathbf{y}}^\top (\mathbf{\Theta}^{(L)})^{-1} \widetilde{\mathbf{y}}}/(m\sqrt{n})$ for some small enough absolute constant $\kappa$ satisfies

$$\mathbb{E}[L_{\mathcal{D}}^{0-1}(\widehat{\mathbf{W}})] \leq \widetilde{\mathcal{O}}\left[ L \cdot \inf_{\widetilde{y}_i y_i \geq 1} \sqrt{\frac{\widetilde{\mathbf{y}}^\top (\mathbf{\Theta}^{(L)})^{-1} \widetilde{\mathbf{y}}}{n}} \right] + \mathcal{O}\left[ \sqrt{\frac{\log(1/\delta)}{n}} \right],$$

where the expectation is taken over the uniform draw of $\widehat{\mathbf{W}}$ from $\{\mathbf{W}^{(0)}, \ldots, \mathbf{W}^{(n-1)}\}$.

**Remark 3.11.** Corollary 3.10 gives an algorithm-dependent generalization error bound of over-parameterized $L$-layer neural networks trained with SGD. It is worth noting that recently Arora et al. [4] gives a generalization bound $\widetilde{\mathcal{O}}(\sqrt{\mathbf{y}^\top (\mathbf{H}^\infty)^{-1} \mathbf{y}/n})$ for two-layer networks with fixed second layer weights, where $\mathbf{H}^\infty$ is defined as

$$\mathbf{H}^\infty_{i,j} = \langle \mathbf{x}_i, \mathbf{x}_j \rangle \cdot \mathbb{E}_{\mathbf{w} \sim N(\mathbf{0}, \mathbf{I})}[\sigma'(\mathbf{w}^\top \mathbf{x}_i) \sigma'(\mathbf{w}^\top \mathbf{x}_j)].$$

Our result in Corollary 3.10 can be specialized to two-layer neural networks by choosing $L = 2$, and yields a bound $\widetilde{\mathcal{O}}(\sqrt{\mathbf{y}^\top (\mathbf{\Theta}^{(2)})^{-1} \mathbf{y}/n})$, where

$$\mathbf{\Theta}^{(2)}_{i,j} = \mathbf{H}^\infty_{i,j} + 2 \cdot \mathbb{E}_{\mathbf{w} \sim N(\mathbf{0}, \mathbf{I})}[\sigma(\mathbf{w}^\top \mathbf{x}_i) \sigma(\mathbf{w}^\top \mathbf{x}_j)].$$

Here the extra term $2 \cdot \mathbb{E}_{\mathbf{w} \sim N(\mathbf{0}, \mathbf{I})}[\sigma(\mathbf{w}^\top \mathbf{x}_i) \sigma(\mathbf{w}^\top \mathbf{x}_j)]$ corresponds to the training of the second layer–it is the limit of $\frac{1}{m}\langle \nabla_{\mathbf{W}_2} f_{\mathbf{W}^{(0)}}(\mathbf{x}_i), \nabla_{\mathbf{W}_2} f_{\mathbf{W}^{(0)}}(\mathbf{x}_j) \rangle$. Since we have $\mathbf{\Theta}^{(2)} \succeq \mathbf{H}^\infty$, our bound is sharper than theirs. This comparison also shows that, our result generalizes the result in Arora et al. [4] from two-layer, fixed second layer networks to deep networks with all parameters being trained.

**Remark 3.12.** Corollary 3.10 is based on the asymptotic convergence result in Lemma 3.8, which does not show how wide the network need to be in order to make the Gram matrix close enough to the NTK matrix. Very recently, Arora et al. [3] provided a non-asymptotic convergence result for the Gram matrix, and showed the equivalence between an infinitely wide network trained by gradient flow and a kernel regression predictor using neural tangent kernel, which suggests that the generalization of deep neural networks trained by gradient flow can potentially be measured by the corresponding NTK. Utilizing this non-asymptotic convergence result, one can potentially specify the detailed dependency of $\widetilde{m}^*(\delta, L, n, \lambda_0)$ on $\delta$, $L$, $n$ and $\lambda_0$ in Corollary 3.10.

**Remark 3.13.** Corollary 3.10 demonstrates that the generalization bound given by Theorem 3.3 does not increase with network width $m$, as long as $m$ is large enough. Moreover, it provides a clear characterization of the classifiability of data. In fact, the $\sqrt{\widetilde{\mathbf{y}}^\top (\mathbf{\Theta}^{(L)})^{-1} \widetilde{\mathbf{y}}}$ factor in the generalization bound given in Corollary 3.10 is exactly the NTK-induced RKHS norm of the kernel regression classifier on data $\{(\mathbf{x}_i, \widetilde{y}_i)\}_{i=1}^n$. Therefore, if $y = f^*(\mathbf{x})$ for some $f^*(\cdot)$ with bounded norm in the NTK-induced reproducing kernel Hilbert space (RKHS), then over-parameterized neural networks trained with SGD generalize well. In Appendix E, we provide some numerical evaluation of the leading terms in the generalization bounds in Theorem 3.3 and Corollary 3.10 to demonstrate that they are very informative on real-world datasets.

# 4 Proof of Main Theory

In this section we provide the proof of Theorem 3.3 and Corollary 3.10, and explain the intuition behind the proof. For notational simplicity, for $i \in [n]$ we denote $L_i(\mathbf{W}) = L_{(\mathbf{x}_i, y_i)}(\mathbf{W})$.

## 4.1 Proof of Theorem 3.3

Before giving the proof of Theorem 3.3, we first introduce several lemmas. The following lemma states that near initialization, the neural network function is almost linear in terms of its weights.

**Lemma 4.1.** There exists an absolute constant $\kappa$ such that, with probability at least $1 - \mathcal{O}(nL^2) \cdot \exp[-\Omega(m\omega^{2/3}L)]$ over the randomness of $\mathbf{W}^{(0)}$, for all $i \in [n]$ and $\mathbf{W}, \mathbf{W}' \in \mathcal{B}(\mathbf{W}^{(0)}, \omega)$ with $\omega \leqslant \kappa L^{-6}[\log(m)]^{-3/2}$, it holds uniformly that

$$|f_{\mathbf{W}'}(\mathbf{x}_i) - f_{\mathbf{W}}(\mathbf{x}_i) - \langle \nabla f_{\mathbf{W}}(\mathbf{x}_i), \mathbf{W}' - \mathbf{W} \rangle| \leqslant \mathcal{O}\left(\omega^{1/3} L^2 \sqrt{m \log(m)}\right) \cdot \sum_{l=1}^{L-1} \|\mathbf{W}'_l - \mathbf{W}_l\|_2.$$

Since the cross-entropy loss $\ell(\cdot)$ is convex, given Lemma 4.1, we can show in the following lemma that near initialization, $L_i(\mathbf{W})$ is also almost a convex function of $\mathbf{W}$ for any $i \in [n]$.

**Lemma 4.2.** There exists an absolute constant $\kappa$ such that, with probability at least $1 - \mathcal{O}(nL^2) \cdot \exp[-\Omega(m\omega^{2/3}L)]$ over the randomness of $\mathbf{W}^{(0)}$, for any $\epsilon > 0$, $i \in [n]$ and $\mathbf{W}, \mathbf{W}' \in \mathcal{B}(\mathbf{W}^{(0)}, \omega)$ with $\omega \leqslant \kappa L^{-6} m^{-3/8}[\log(m)]^{-3/2}\epsilon^{3/4}$, it holds uniformly that

$$L_i(\mathbf{W}') \geqslant L_i(\mathbf{W}) + \langle \nabla_{\mathbf{W}} L_i(\mathbf{W}), \mathbf{W}' - \mathbf{W} \rangle - \epsilon.$$

The locally almost convex property of the loss function given by Lemma 4.2 implies that the dynamics of Algorithm 1 is similar to the dynamics of convex optimization. We can therefore derive a bound of the cumulative loss. The result is given in the following lemma.

**Lemma 4.3.** For any $\epsilon, \delta, R > 0$, there exists

$$m^*(\epsilon, \delta, R, L) = \widetilde{\mathcal{O}}\big(\text{poly}(R, L)\big) \cdot \epsilon^{-14} \cdot \log(1/\delta)$$

such that if $m \geqslant m^*(\epsilon, \delta, R, L)$, then with probability at least $1 - \delta$ over the randomness of $\mathbf{W}^{(0)}$, for any $\mathbf{W}^* \in \mathcal{B}(\mathbf{W}^{(0)}, Rm^{-1/2})$, Algorithm 1 with $\eta = \nu\epsilon/(Lm)$, $n = L^2 R^2/(2\nu\epsilon^2)$ for some small enough absolute constant $\nu$ has the following cumulative loss bound:

$$\sum_{i=1}^n L_i(\mathbf{W}^{(i-1)}) \leqslant \sum_{i=1}^n L_i(\mathbf{W}^*) + 3n\epsilon.$$

We now finalize the proof by applying an online-to-batch conversion argument [9], and use Lemma 4.1 to relate the neural network function with a function in the NTRF function class.

*Proof of Theorem 3.3.* For $i \in [n]$, let $L_i^{0-1}(\mathbf{W}^{(i-1)}) = \mathbb{1}\{y_i \cdot f_{\mathbf{W}^{(i-1)}}(\mathbf{x}_i) < 0\}$. Since cross-entropy loss satisfies $\mathbb{1}\{z \leq 0\} \leq 4\ell(z)$, we have $L_i^{0-1}(\mathbf{W}^{(i-1)}) \leq 4L_i(\mathbf{W}^{(i-1)})$. Therefore, setting $\epsilon = LR/\sqrt{2\nu n}$ in Lemma 4.3 gives that, if $\eta$ is set as $\sqrt{\nu/2}R/(m\sqrt{n})$, then with probability at least $1 - \delta$,

$$\frac{1}{n}\sum_{i=1}^{n} L_i^{0-1}(\mathbf{W}^{(i-1)}) \leq \frac{4}{n}\sum_{i=1}^{n} L_i(\mathbf{W}^*) + \frac{12}{\sqrt{2\nu}} \cdot \frac{LR}{\sqrt{n}}. \tag{4.1}$$

Note that for any $i \in [n]$, $\mathbf{W}^{(i-1)}$ only depends on $(\mathbf{x}_1, y_1), \ldots, (\mathbf{x}_{i-1}, y_{i-1})$ and is independent of $(\mathbf{x}_i, y_i)$. Therefore by Proposition 1 in Cesa-Bianchi et al. [9], with probability at least $1 - \delta$ we have

$$\frac{1}{n}\sum_{i=1}^{n} L_{\mathcal{D}}^{0-1}(\mathbf{W}^{(i-1)}) \leq \frac{1}{n}\sum_{i=1}^{n} L_i^{0-1}(\mathbf{W}^{(i-1)}) + \sqrt{\frac{2\log(1/\delta)}{n}}. \tag{4.2}$$

By definition, we have $\frac{1}{n}\sum_{i=1}^{n} L_{\mathcal{D}}^{0-1}(\mathbf{W}^{(i-1)}) = \mathbb{E}[L_{\mathcal{D}}^{0-1}(\widehat{\mathbf{W}})]$. Therefore combining (4.1) and (4.2) and applying union bound, we obtain that with probability at least $1 - 2\delta$,

$$\mathbb{E}[L_{\mathcal{D}}^{0-1}(\widehat{\mathbf{W}})] \leq \frac{4}{n}\sum_{i=1}^{n} L_i(\mathbf{W}^*) + \frac{12}{\sqrt{2\nu}} \cdot \frac{LR}{\sqrt{n}} + \sqrt{\frac{2\log(1/\delta)}{n}} \tag{4.3}$$

for all $\mathbf{W}^* \in \mathcal{B}(\mathbf{W}^{(0)}, Rm^{-1/2})$. We now compare the neural network function $f_{\mathbf{W}^*}(\mathbf{x}_i)$ with the function $F_{\mathbf{W}^{(0)}, \mathbf{W}^*}(\mathbf{x}_i) := f_{\mathbf{W}^{(0)}}(\mathbf{x}_i) + \langle \nabla f_{\mathbf{W}^{(0)}}(\mathbf{x}_i), \mathbf{W}^* - \mathbf{W}^{(0)} \rangle \in \mathcal{F}(\mathbf{W}^{(0)}, R)$. We have

$$L_i(\mathbf{W}^*) \leq \ell[y_i \cdot F_{\mathbf{W}^{(0)}, \mathbf{W}^*}(\mathbf{x}_i)] + \mathcal{O}\Big((Rm^{-1/2})^{1/3}L^2\sqrt{m\log(m)}\Big) \cdot \sum_{l=1}^{L-1}\|\mathbf{W}_l^* - \mathbf{W}_l^{(0)}\|_2$$

$$\leq \ell[y_i \cdot F_{\mathbf{W}^{(0)}, \mathbf{W}^*}(\mathbf{x}_i)] + \mathcal{O}\Big(L^3\sqrt{m\log(m)}\Big) \cdot R^{4/3} \cdot m^{-2/3}$$

$$\leq \ell[y_i \cdot F_{\mathbf{W}^{(0)}, \mathbf{W}^*}(\mathbf{x}_i)] + LRn^{-1/2},$$

where the first inequality is by the 1-Lipschitz continuity of $\ell(\cdot)$ and Lemma 4.1, the second inequality is by $\mathbf{W}^* \in \mathcal{B}(\mathbf{W}^{(0)}, Rm^{-1/2})$, and last inequality holds as long as $m \geq C_1 R^2 L^{12}[\log(m)]^3 n^3$ for some large enough absolute constant $C_1$. Plugging the inequality above into (4.3) gives

$$\mathbb{E}[L_{\mathcal{D}}^{0-1}(\widehat{\mathbf{W}})] \leq \frac{4}{n}\sum_{i=1}^{n} \ell[y_i \cdot F_{\mathbf{W}^{(0)}, \mathbf{W}^*}(\mathbf{x}_i)] + \left(1 + \frac{12}{\sqrt{2\nu}}\right) \cdot \frac{LR}{\sqrt{n}} + \sqrt{\frac{2\log(1/\delta)}{n}}.$$

Taking infimum over $\mathbf{W}^* \in \mathcal{B}(\mathbf{W}^{(0)}, Rm^{-1/2})$ and rescaling $\delta$ finishes the proof. $\qquad\square$

## 4.2 Proof of Corollary 3.10

In this subsection we prove Corollary 3.10. The following lemma shows that at initialization, with high probability, the neural network function value at all the training inputs are of order $\widetilde{\mathcal{O}}(1)$.

**Lemma 4.4.** For any $\delta > 0$, if $m \geq KL\log(nL/\delta)$ for a large enough absolute constant $K$, then with probability at least $1 - \delta$, $|f_{\mathbf{W}^{(0)}}(\boldsymbol{x}_i)| \leq \mathcal{O}(\sqrt{\log(n/\delta)})$ for all $i \in [n]$.

We now present the proof of Corollary 3.10. The idea is to construct suitable target values $\widehat{y}_1, \ldots, \widehat{y}_n$, and then bound the norm of the solution of the linear equations $\widehat{y}_i = \langle \nabla f_{\mathbf{W}^{(0)}}(\mathbf{x}_i), \mathbf{W} \rangle$, $i \in [n]$. In specific, for any $\widetilde{\mathbf{y}}$ with $\widetilde{y}_i y_i \geq 1$, we examine the *minimum distance solution* to $\mathbf{W}^{(0)}$ that fit the data $\{(\mathbf{x}_i, \widetilde{y}_i)\}_{i=1}^{n}$ well and use it to construct a specific function in $\mathcal{F}(\mathbf{W}^{(0)}, \widetilde{\mathcal{O}}(\sqrt{\widetilde{\mathbf{y}}^\top(\boldsymbol{\Theta}^{(L)})^{-1}\widetilde{\mathbf{y}}}))$.

*Proof of Corollary 3.10.* Set $B = \log\{1/[\exp(n^{-1/2}) - 1]\} = \mathcal{O}(\log(n))$, then for cross-entropy loss we have $\ell(z) \leq n^{-1/2}$ for $z \geq B$. Moreover, let $B' = \max_{i\in[n]} |f_{\mathbf{W}^{(0)}}(\boldsymbol{x}_i)|$. Then by Lemma 4.4, with probability at least $1 - \delta$, $B' \leq \mathcal{O}(\sqrt{\log(n/\delta)})$ for all $i \in [n]$. For any $\widetilde{y}$ with $\widetilde{y}_i y_i \geq 1$, let $\overline{B} = B + B'$ and $\widehat{\mathbf{y}} = \overline{B} \cdot \widetilde{\mathbf{y}}$, then it holds that for any $i \in [n]$,

$$y_i \cdot [\widehat{y}_i + f_{\mathbf{W}^{(0)}}(\boldsymbol{x}_i)] = y_i \cdot \widehat{y}_i + y_i \cdot f_{\mathbf{W}^{(0)}}(\boldsymbol{x}_i) \geq B + B' - B' \geq B,$$

and therefore

$$\ell\{y_i \cdot [\widehat{y}_i + f_{\mathbf{W}^{(0)}}(\boldsymbol{x}_i)]\} \leqslant n^{-1/2}, \ i \in [n]. \tag{4.4}$$

Denote $\mathbf{F} = m^{-1/2} \cdot (\mathrm{vec}[\nabla f_{\mathbf{W}^{(0)}}(\mathbf{x}_1)], \ldots, \mathrm{vec}[\nabla f_{\mathbf{W}^{(0)}}(\mathbf{x}_n)]) \in \mathbb{R}^{[md+m+m^2(L-2)]\times n}$. Note that entries of $\boldsymbol{\Theta}^{(L)}$ are all bounded by $L$. Therefore, the largest eigenvalue of $\boldsymbol{\Theta}^{(L)}$ is at most $nL$, and we have $\widetilde{\mathbf{y}}^\top (\boldsymbol{\Theta}^{(L)})^{-1}\widetilde{\mathbf{y}} \geqslant n^{-1}L^{-1}\|\widetilde{\mathbf{y}}\|_2^2 = L^{-1}$. By Lemma 3.8 and standard matrix perturbation bound, there exists $m^*(\delta, L, n, \lambda_0)$ such that, if $m \geqslant m^*(\delta, L, n, \lambda_0)$, then with probability at least $1 - \delta$, $\mathbf{F}^\top\mathbf{F}$ is strictly positive-definite and

$$\|(\mathbf{F}^\top\mathbf{F})^{-1} - (\boldsymbol{\Theta}^{(L)})^{-1}\|_2 \leqslant \inf_{\widetilde{y}_i y_i \geqslant 1} \widetilde{\mathbf{y}}^\top (\boldsymbol{\Theta}^{(L)})^{-1}\widetilde{\mathbf{y}}/n. \tag{4.5}$$

Let $\mathbf{F} = \mathbf{P}\boldsymbol{\Lambda}\mathbf{Q}^\top$ be the singular value decomposition of $\mathbf{F}$, where $\mathbf{P} \in \mathbb{R}^{m \times n}, \mathbf{Q} \in \mathbb{R}^{n \times n}$ have orthogonal columns, and $\boldsymbol{\Lambda} \in \mathbb{R}^{n \times n}$ is a diagonal matrix. Let $\mathbf{w}_{\mathrm{vec}} = \mathbf{P}\boldsymbol{\Lambda}^{-1}\mathbf{Q}^\top\widehat{\mathbf{y}}$, then we have

$$\mathbf{F}^\top\mathbf{w}_{\mathrm{vec}} = (\mathbf{Q}\boldsymbol{\Lambda}\mathbf{P}^\top)(\mathbf{P}\boldsymbol{\Lambda}^{-1}\mathbf{Q}^\top\widehat{\mathbf{y}}) = \widehat{\mathbf{y}}. \tag{4.6}$$

Moreover, by direct calculation we have

$$\|\mathbf{w}_{\mathrm{vec}}\|_2^2 = \|\mathbf{P}\boldsymbol{\Lambda}^{-1}\mathbf{Q}^\top\widehat{\mathbf{y}}\|_2^2 = \|\boldsymbol{\Lambda}^{-1}\mathbf{Q}^\top\widehat{\mathbf{y}}\|_2^2 = \widehat{\mathbf{y}}^\top\mathbf{Q}\boldsymbol{\Lambda}^{-2}\mathbf{Q}^\top\widehat{\mathbf{y}} = \widehat{\mathbf{y}}^\top(\mathbf{F}^\top\mathbf{F})^{-1}\widehat{\mathbf{y}}.$$

Therefore by (4.5) and the fact that $\|\widehat{\mathbf{y}}\|_2^2 = \overline{B}^2 n$, we have

$$\begin{aligned}
\|\mathbf{w}_{\mathrm{vec}}\|_2^2 &= \widehat{\mathbf{y}}^\top[(\mathbf{F}^\top\mathbf{F})^{-1} - (\boldsymbol{\Theta}^{(L)})^{-1}]\widehat{\mathbf{y}} + \widehat{\mathbf{y}}^\top(\boldsymbol{\Theta}^{(L)})^{-1}\widehat{\mathbf{y}} \\
&\leqslant \overline{B}^2 \cdot n \cdot \|(\mathbf{F}^\top\mathbf{F})^{-1} - (\boldsymbol{\Theta}^{(L)})^{-1}\|_2 + \overline{B}^2 \cdot \widetilde{\mathbf{y}}^\top(\boldsymbol{\Theta}^{(L)})^{-1}\widetilde{\mathbf{y}} \\
&\leqslant 2\overline{B}^2 \cdot \widetilde{\mathbf{y}}^\top(\boldsymbol{\Theta}^{(L)})^{-1}\widetilde{\mathbf{y}}.
\end{aligned}$$

Let $\mathbf{W} \in \mathcal{W}$ be the parameter collection reshaped from $m^{-1/2}\mathbf{w}_{\mathrm{vec}}$. Then clearly

$$\|\mathbf{W}_l\|_F \leqslant m^{-1/2}\|\mathbf{w}_{\mathrm{vec}}\|_2 \leqslant \widetilde{\mathcal{O}}\Big(\sqrt{\widetilde{\mathbf{y}}^\top(\boldsymbol{\Theta}^{(L)})^{-1}\widetilde{\mathbf{y}}} \cdot m^{-1/2}\Big),$$

and therefore $\mathbf{W} \in \mathcal{B}\big(\mathbf{0}, \mathcal{O}\big(\sqrt{\widetilde{\mathbf{y}}^\top(\boldsymbol{\Theta}^{(L)})^{-1}\widetilde{\mathbf{y}}} \cdot m^{-1/2}\big)\big)$. Moreover, by (4.6), we have $\widehat{y}_i = \langle\nabla_{\mathbf{W}}f_{\mathbf{W}^{(0)}}(\mathbf{x}_i), \mathbf{W}\rangle$. Plugging this into (4.4) then gives

$$\ell\big\{y_i \cdot \big[f_{\mathbf{W}^{(0)}}(\mathbf{x}_i) + \langle\nabla_{\mathbf{W}}f_{\mathbf{W}^{(0)}}(\boldsymbol{x}_i), \mathbf{W}\rangle\big]\big\} \leqslant n^{-1/2}.$$

Since $\widehat{f}(\cdot) = f_{\mathbf{W}^{(0)}}(\cdot) + \langle\nabla_{\mathbf{W}}f_{\mathbf{W}^{(0)}}(\cdot), \mathbf{W}\rangle \in \mathcal{F}\big(\mathbf{W}^{(0)}, \widetilde{\mathcal{O}}\big(\sqrt{\widetilde{\mathbf{y}}^\top(\boldsymbol{\Theta}^{(L)})^{-1}\widetilde{\mathbf{y}}}\big)\big)$, applying Theorem 3.3 and taking infimum over $\widetilde{\mathbf{y}}$ completes the proof. □

# 5    Conclusions and Future Work

In this paper we provide an expected 0-1 error bound for wide and deep ReLU networks trained with SGD. This generalization error bound is measured by the NTRF function class. The connection to the neural tangent kernel function studied in Jacot et al. [18] is also discussed. Our result covers a series of recent generalization bounds for wide enough neural networks, and provides better bounds.

An important future work is to improve the over-parameterization condition in Theorem 3.3 and Corollary 3.10. Other future directions include proving sample complexity lower bounds in the over-parameterized regime, implementing the results in Jain et al. [19] to obtain last iterate bound of SGD, and establishing uniform convergence based generalization bounds for over-parameterized neural networks with methods developped in Bartlett et al. [6], Neyshabur et al. [27], Long and Sedghi [26].

## Acknowledgement

We would like to thank Peter Bartlett for a valuable discussion, and Simon S. Du for pointing out a related work [3]. We also thank the anonymous reviewers and area chair for their helpful comments. This research was sponsored in part by the National Science Foundation CAREER Award IIS-1906169, IIS-1903202, and Salesforce Deep Learning Research Award. The views and conclusions contained in this paper are those of the authors and should not be interpreted as representing any funding agencies.

## Footnotes

[1]Since random feature models and kernel methods are highly related [31, 32], we group them into the same category. More details are discussed in Section 3.2.

[2]Normalizing weights to the same scale is necessary for a proper comparison. See Appendix A.2 for details.

[3]The original result by Jacot et al. [18] requires that the widths of different layers go to infinity sequentially. Their result was later improved by Yang [36] such that the widths of different layers can go to infinity simultaneously.

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
