[Supplementary Material]

# A Comparison with Recent Results

In this section we compare our result in Theorem 3.3 with recent generalization error bounds for over-paramerized neural networks by Cao and Gu [8], Yehudai and Shamir [37], E et al. [14], and backup our discussions in Remark 3.5 and Remark 3.6.

## A.1 Comparison with Cao and Gu [8]

In this section we provide direct comparison between our result in Theorem 3.3 and Theorem 4.4 in Cao and Gu [8]. To concretely compare these two results, we apply our result to the setting studied in Cao and Gu [8], which is based on the following assumption.

**Assumption A.1.** There exist a constant $\gamma > 0$ and

$$f(\cdot) \in \left\{ f(\mathbf{x}) = \int_{\mathbb{R}^d} c(\mathbf{u})\sigma(\mathbf{u}^\top\mathbf{x})p(\mathbf{u})\mathrm{d}\mathbf{u} : \|c(\cdot)\|_\infty \leqslant 1 \right\},$$

where $p(\mathbf{u})$ the density of standard Gaussian vectors, such that $y \cdot f(\mathbf{x}) \geqslant \gamma$ for all $(\mathbf{x}, y) \in \mathrm{supp}(\mathcal{D})$.

Under Assumption 3.1 and Assumption A.1, in order to train the network to achieve $\epsilon$ expected 0-1 loss, Cao and Gu [8] gave a sample complexity of order $\widetilde{\mathcal{O}}(\mathrm{poly}(2^L, \gamma^{-1}) \cdot \epsilon^{-4})$. In comparison, our result in Theorem 3.3 leads to the following corollary.

**Corollary A.2.** Under Assumption 3.1 and Assumption A.1, for any $\delta \in (0, e^{-1}]$, there exists

$$m^*(\delta, \gamma, L, n) = \widetilde{\mathcal{O}}\big(\mathrm{poly}(2^L, \gamma^{-1})\big) \cdot n^7 \cdot \log(1/\delta)$$

such that if $m \geqslant m^*(\delta, R, L, n)$, then with probability at least $1 - \delta$ over the randomness of $\mathbf{W}^{(0)}$, the parameters given by Algorithm 1 with $\eta = \kappa \cdot R/(m\sqrt{n})$ for some small enough absolute constant $\kappa$ satisfies

$$\mathbb{E}\big[L_\mathcal{D}^{0-1}(\widehat{\mathbf{W}})\big] \leqslant \widetilde{\mathcal{O}}\left(\frac{2^L \cdot \gamma^{-1}}{\sqrt{n}}\right),$$

where the expectation is taken over the draws of training examples $\{(\mathbf{x}_i, y_i)\}_{i=1}^n$ as well as the uniform draw of $\widehat{\mathbf{W}}$ from $\{\mathbf{W}^{(0)}, \ldots, \mathbf{W}^{(n-1)}\}$.

By setting the expected 0-1 loss bound to $\epsilon$, we obtain a sample complexity of order $\widetilde{\mathcal{O}}(4^L \cdot \gamma^{-2}\epsilon^{-2})$, which is better than the sample complexity given in Cao and Gu [8] by a factor of $\epsilon^{-2}$.

## A.2 Comparison with Yehudai and Shamir [37], E et al. [14]

Here we give a detailed explanation to Remark 3.6, where we compare our result with Yehudai and Shamir [37], E et al. [14]. The reference function classes studied in these two papers share the same general form:

$$\big\{f(x) = \mathbf{W}_2\sigma(\mathbf{W}_1^{(0)}\mathbf{x}) : \|\mathbf{W}_2\|_F \leqslant Cm^{-1/2}\big\},$$

where $C$ is a constant, and $\mathbf{W}_1^{(0)} \in \mathbb{R}^{m \times d}$ is the first layer parameter matrix whose rows are sampled from certain distribution $\pi$ associated to the initialization scheme. Specifically, Yehudai and Shamir [37] studied the case where $\pi$ is the uniform distribution over the $d$-dimensional cube $[-d^{-1/2}, d^{-1/2}]^d$, while E et al. [14] studied the uniform distribution over the sphere $S^{d-1}$. By standard concentration inequality, we can see that in both papers, with high probability, the distribution $\pi$ gives $\mathbf{W}_1^{(0)}$ with $\|\mathbf{W}_1^{(0)}\|_2 \approx \mathcal{O}(m^{1/2})$. In terms of second layer initialization $\mathbf{W}_2^{(0)}$, the generalization results in both papers require that $\|\mathbf{W}_2^{(0)}\|_2 \leqslant \mathcal{O}(m^{-1/2})$. With such a scaling, we can apply the following lemma.

**Lemma A.3.** Suppose that $\mathbf{W}^{(0)} = (\mathbf{W}_1^{(0)}, \mathbf{W}_2^{(0)}) \in \mathbb{R}^{m \times d} \times \mathbb{R}^{1 \times m}$ be weights satisfying $\|\mathbf{W}_2^{(0)}\|_F \leqslant Km^{-1/2}$ for some $K = \widetilde{\mathcal{O}}(1)$, then

$$\big\{f(x) = \mathbf{W}_2\sigma(\mathbf{W}_1^{(0)}\mathbf{x}) : \|\mathbf{W}_2\|_F \leqslant Cm^{-1/2}\big\} \subseteq \mathcal{F},$$

where

$$\mathcal{F} = \big\{\mathbf{W}_2^{(0)}\sigma(\mathbf{W}_1^{(0)}\mathbf{x}) + \mathbf{W}_2\sigma(\mathbf{W}_1^{(0)}\mathbf{x}) : \|\mathbf{W}_2\|_F \leqslant (C + K) \cdot m^{-1/2}\big\},$$

and $\sigma(\cdot)$ is the activation function of interest.

We compare our result with the bounds given by Yehudai and Shamir [37], E et al. [14] by comparing the reference function classes we use. Apparently, a larger reference function class in general gives a better generalization error bound. Such a comparison requires us to adjust the scaling of initialized parameters. Based on our previous discussion, it is easy to see that the initialized second layer weights in our work and Yehudai and Shamir [37], E et al. [14] are all of the same scaling. However, the $\|\cdot\|_2$ of first layer weight matrix in Yehudai and Shamir [37], E et al. [14] is larger than ours by a factor of $\sqrt{m}$. Adjusting this scaling difference will give an extra factor $\sqrt{m}$, which matches the $\sqrt{m}$ factor in the definition of our neural network function. Note that even after adjusting the scaling of parameters, these random feature function classes are not directly comparable, since the activation functions and the distributions of random weights are different. However, an informal comparison can already clearly show the advantage of our result. Moreover, we remark that at least for two-layer networks, our analysis can be easily generalized to other activation functions and initialization methods, and the resulting NTRF class should be strictly larger than the random feature function classes used in Yehudai and Shamir [37], E et al. [14]. This justifies our discussion in Remark 3.6.

# B Proofs of Technical Lemmas in Section 4

In this section we provide the proofs of the technical lemmas in Section 4. We first introduce some extra notations. Following Allen-Zhu et al. [2], for a parameter collection $\mathbf{W}$ and $i \in [n]$, we denote

$$\mathbf{h}_{i,0} = \mathbf{x}_i, \ \mathbf{h}_{i,l} = \sigma(\mathbf{W}_l \mathbf{h}_{i,l-1}), l \in [L-1]$$

as the hidden layer outputs of the network. We also define binary diagonal matrices

$$\mathbf{D}_{i,l} = \mathrm{diag}\big(\mathbb{1}\{(\mathbf{W}_l \mathbf{h}_{i,l})_1 > 0\}, \dots, \mathbb{1}\{(\mathbf{W}_l \mathbf{h}_{i,l})_m > 0\}\big), l \in [L-1].$$

For $i \in [n]$ and $l \in [L-1]$, we use $\mathbf{h}'_{i,l}, \mathbf{D}'_{i,l}$ and $\mathbf{h}^{(0)}_{i,l}, \mathbf{D}^{(0)}_{i,l}$ to denote the hidden layer outputs and binary diagonal matrices with parameter collections $\mathbf{W}'$ and $\mathbf{W}^{(0)}$ respectively. We also implement the following matrix product notation which is also used in Zou et al. [39], Cao and Gu [8]:

$$\prod_{r=l_1}^{l_2} \mathbf{A}_r := \begin{cases} \mathbf{A}_{l_2} \mathbf{A}_{l_2-1} \cdots \mathbf{A}_{l_1} & \text{if } l_1 \leqslant l_2 \\ \mathbf{I} & \text{otherwise.} \end{cases}$$

With this notation, we have the following matrix product representation of the neural network gradients:

$$\nabla_{\mathbf{W}_l} f_{\mathbf{W}}(\mathbf{x}_i) = \begin{cases} \sqrt{m} \cdot \big[\mathbf{h}_{i,l-1} \mathbf{W}_L \big(\prod_{r=l+1}^{L-1} \mathbf{D}_{i,r} \mathbf{W}_r\big) \mathbf{D}_{i,l}\big]^\top, & l \in [L-1], \\ \sqrt{m} \cdot \mathbf{h}_{i,L-1}^\top, & l = L. \end{cases}$$

## B.1 Proof of Lemma 4.1

The following two lemmas are proved based on several results given by Allen-Zhu et al. [2]. Note that in their paper, both the first and the last layers of the network are fixed, which is slightly different from our setting. We remark that this difference does not affect the result.

**Lemma B.1.** If $\omega \leqslant \mathcal{O}(L^{-9/2}[\log(m)]^{-3})$, then with probability at least $1 - \mathcal{O}(nL) \cdot \exp[-\Omega(m\omega^{2/3}L)]$, $1/2 \leqslant \|\mathbf{h}_{i,l}\|_2 \leqslant 3/2$ for all $\mathbf{W} \in \mathcal{B}(\mathbf{W}^{(0)}, \omega)$, $i \in [n]$ and $l \in [L-1]$.

**Lemma B.2.** If $\omega \leqslant \mathcal{O}(L^{-6}[\log(m)]^{-3})$, then with probability at least $1 - \mathcal{O}(nL^2) \cdot \exp[-\Omega(m\omega^{2/3}L)]$, uniformly over:

- any $i \in [n]$, $1 \leqslant l_1 < l_2 \leqslant L-1$

- any diagonal matrices $\mathbf{D}''_{i,1}, \dots, \mathbf{D}''_{i,L-1} \in [-1,1]^{m \times m}$ with at most $\mathcal{O}(m\omega^{2/3}L)$ non-zero entries,

the following results hold:

(i) For all $\mathbf{W} \in \mathcal{B}(\mathbf{W}^{(0)}, \omega)$, $\|\prod_{r=l_1}^{l_2}(\mathbf{D}_{i,r} + \mathbf{D}''_{i,r})\mathbf{W}_r\|_2 \leqslant \mathcal{O}(\sqrt{L})$.

(ii) For all $\mathbf{W} \in \mathcal{B}(\mathbf{W}^{(0)}, \omega)$, $\|\mathbf{W}_L \prod_{r=l_1}^{L-1}(\mathbf{D}_{i,r} + \mathbf{D}''_{i,r})\mathbf{W}_r\|_2 \leqslant \mathcal{O}(1)$.

(iii) For all $\mathbf{W}, \mathbf{W}' \in \mathcal{B}(\mathbf{W}^{(0)}, \omega)$,

$$\left\| \mathbf{W}'_L \prod_{r=l_1}^{L-1} (\mathbf{D}'_{i,r} + \mathbf{D}''_{i,r}) \mathbf{W}'_r - \mathbf{W}_L \prod_{r=l_1}^{L-1} \mathbf{D}_{i,r} \mathbf{W}_r \right\|_2 \leqslant \mathcal{O}\left( \omega^{1/3} L^2 \sqrt{\log(m)} \right).$$

We are now ready to prove Lemma 4.1.

*Proof of Lemma 4.1.* Since $f_{\mathbf{W}'}(\mathbf{x}_i) = \sqrt{m} \cdot \mathbf{W}'_L \mathbf{h}'_{i,L-1}$, $f_{\mathbf{W}}(\mathbf{x}_i) = \sqrt{m} \cdot \mathbf{W}_L \mathbf{h}_{i,L-1}$, by direct calculation, we have

$$f_{\mathbf{W}'}(\mathbf{x}_i) - F_{\mathbf{W},\mathbf{W}'}(\mathbf{x}_i) = -\sqrt{m} \cdot \sum_{l=1}^{L-1} \mathbf{W}_L \left( \prod_{r=l+1}^{L-1} \mathbf{D}_{i,r} \mathbf{W}_r \right) \mathbf{D}_{i,l} (\mathbf{W}'_l - \mathbf{W}_l) \mathbf{h}_{i,l-1}$$
$$+ \sqrt{m} \cdot \mathbf{W}'_L (\mathbf{h}'_{i,L-1} - \mathbf{h}_{i,L-1}).$$

By Claim 8.2 in Allen-Zhu et al. [2] , there exist diagonal matrices $\mathbf{D}''_{i,l} \in \mathbb{R}^{m \times m}$ with entries in $[-1,1]$ such that $\|\mathbf{D}''_{i,l}\|_0 \leqslant \mathcal{O}(m \omega^{2/3} L)$ and

$$\mathbf{h}_{i,L-1} - \mathbf{h}'_{i,L-1} = \sum_{l=1}^{L-1} \left[ \prod_{r=l+1}^{L-1} (\mathbf{D}'_{i,r} + \mathbf{D}''_{i,r}) \mathbf{W}'_r \right] (\mathbf{D}'_{i,l} + \mathbf{D}''_{i,l})(\mathbf{W}_l - \mathbf{W}'_l) \mathbf{h}_{i,l-1}$$

for all $i \in [n]$. Therefore

$$f_{\mathbf{W}'}(\mathbf{x}_i) - F_{\mathbf{W},\mathbf{W}'}(\mathbf{x}_i) = \sqrt{m} \cdot \sum_{l=1}^{L-1} \mathbf{W}'_L \left[ \prod_{r=l+1}^{L-1} (\mathbf{D}'_{i,r} + \mathbf{D}''_{i,r}) \mathbf{W}'_r \right] (\mathbf{D}'_{i,l} + \mathbf{D}''_{i,l})(\mathbf{W}_l - \mathbf{W}'_l) \mathbf{h}_{i,l-1}$$
$$- \sqrt{m} \cdot \sum_{l=1}^{L-1} \mathbf{W}_L \left( \prod_{r=l+1}^{L-1} \mathbf{D}_{i,r} \mathbf{W}_r \right) \mathbf{D}_{i,l} (\mathbf{W}'_l - \mathbf{W}_l) \mathbf{h}_{i,l-1}.$$

By (iii) in Lemma B.2, with probability at least $1 - \mathcal{O}(nL^2) \cdot \exp[-\Omega(m \omega^{2/3} L)]$, we have

$$|f_{\mathbf{W}'}(\mathbf{x}_i) - F_{\mathbf{W},\mathbf{W}'}(\mathbf{x}_i)| \leqslant \mathcal{O}\left( \omega^{1/3} L^2 \sqrt{m \log(m)} \right) \cdot \sum_{l=1}^{L-1} \|\mathbf{h}_{i,l-1}\|_2 \cdot \|\mathbf{W}'_l - \mathbf{W}_l\|_2$$
$$\leqslant \mathcal{O}\left( \omega^{1/3} L^2 \sqrt{m \log(m)} \right) \cdot \sum_{l=1}^{L-1} \|\mathbf{W}'_l - \mathbf{W}_l\|_2,$$

where the last inequality follows by Lemma B.1. This inequality finishes the proof. $\qquad \square$

## B.2    Proof of Lemma 4.2

Intuitively, Lemma 4.2 follows by the fact that the composition of a convex function and an almost linear function is almost convex. The detailed proof is as follows.

*Proof of Lemma 4.2.* By the convexity of $\ell(z)$, we have

$$L_i(\mathbf{W}') - L_i(\mathbf{W}) = \ell[y_i f_{\mathbf{W}'}(\mathbf{x}_i)] - \ell[y_i f_{\mathbf{W}}(\mathbf{x}_i)] \geqslant \ell'[y_i f_{\mathbf{W}}(\mathbf{x}_i)] \cdot y_i \cdot [f_{\mathbf{W}'}(\mathbf{x}_i) - f_{\mathbf{W}}(\mathbf{x}_i)].$$

By chain rule, we have

$$\sum_{l=1}^{L} \langle \nabla_{\mathbf{W}_l} L_i(\mathbf{W}), \mathbf{W}'_l - \mathbf{W}_l \rangle = \ell'[y_i f_{\mathbf{W}}(\mathbf{x}_i)] \cdot y_i \cdot \langle \nabla f_{\mathbf{W}}(\mathbf{x}_i), \mathbf{W}' - \mathbf{W} \rangle.$$

Therefore by triangle inequality, we have

$$\ell'[y_i f_{\mathbf{W}}(\mathbf{x}_i)] \cdot y_i \cdot [f_{\mathbf{W}'}(\mathbf{x}_i) - f_{\mathbf{W}}(\mathbf{x}_i)] \geqslant \ell'[y_i f_{\mathbf{W}}(\mathbf{x}_i)] \cdot y_i \cdot \langle \nabla f_{\mathbf{W}}(\mathbf{x}_i), \mathbf{W}' - \mathbf{W} \rangle - I$$
$$= \sum_{l=1}^{L} \langle \nabla_{\mathbf{W}_l} L_i(\mathbf{W}), \mathbf{W}'_l - \mathbf{W}_l \rangle - I,$$

where $I = \left| \ell'[y_i f_{\mathbf{W}}(\mathbf{x}_i)] \cdot y_i \cdot \left[ f_{\mathbf{W}'}(\mathbf{x}_i) - f_{\mathbf{W}}(\mathbf{x}_i) - \langle \nabla f_{\mathbf{W}}(\mathbf{x}_i), \mathbf{W}' - \mathbf{W} \rangle \right] \right|$. Then by upper-bounding $I$ with Lemma 4.1 and the fact that $|\ell'[y_i f_{\mathbf{W}}(\mathbf{x}_i)] \cdot y_i| \leqslant 1$, we have

$$L_i(\mathbf{W}') - L_i(\mathbf{W}) \geqslant \sum_{l=1}^{L} \langle \nabla_{\mathbf{W}_l} L_i(\mathbf{W}), \mathbf{W}'_l - \mathbf{W}_l \rangle - \mathcal{O}\left( \omega^{1/3} L^2 \sqrt{m \log(m)} \right) \sum_{l=1}^{L-1} \| \mathbf{W}'_l - \mathbf{W}_l \|_2$$

$$\geqslant \sum_{l=1}^{L} \langle \nabla_{\mathbf{W}_l} L_i(\mathbf{W}), \mathbf{W}'_l - \mathbf{W}_l \rangle - \epsilon,$$

where the last inequality again follows by $\omega \leqslant \mathcal{O}\left( L^{-9/4} m^{-3/8} [\log(m)]^{-3/8} \epsilon^{3/4} \right)$. $\qquad\square$

## B.3    Proof of Lemma 4.3

To prove Lemma 4.3, we first introduce the following lemma which provides an upper bound for the gradient of the neural network function near initialization.

**Lemma B.3.** There exists an absolute constant $\kappa$ such that, with probability at least $1 - \mathcal{O}(nL^2) \cdot \exp[-\Omega(m\omega^{2/3}L)]$, for all $i \in [n]$, $l \in [L]$ and $\mathbf{W} \in \mathcal{B}(\mathbf{W}^{(0)}, \omega)$ with $\omega \leqslant \kappa L^{-6} [\log(m)]^{-3}$, it holds uniformly that

$$\| \nabla_{\mathbf{W}_l} f_{\mathbf{W}}(\mathbf{x}_i) \|_F, \| \nabla_{\mathbf{W}_l} L_i(\mathbf{W}) \|_F \leqslant \mathcal{O}(\sqrt{m}).$$

We now provide the final proof of Lemma 4.3.

*Proof of Lemma 4.3.* Let $\omega = C_1 L^{-6} m^{-3/8} [\log(m)]^{-3} \epsilon^{3/4}$, where $C_1$ is a small enough absolute constant such that the conditions on $\omega$ given in Lemmas 4.2 and B.3 hold. It is easy to see that as long as $m \geqslant C_1^{-8} R^8 L^{48} [\log(m)]^{12} \epsilon^{-6}$, we have $\mathbf{W}^* \in \mathcal{B}(\mathbf{W}^{(0)}, \omega)$. We now show that under our parameter choice, $\mathbf{W}^{(0)}, \ldots, \mathbf{W}^{(n-1)}$ are inside $\mathcal{B}(\mathbf{W}^{(0)}, \omega)$ as well.

This result follows by simple induction. Clearly we have $\mathbf{W}^{(0)} \in \mathcal{B}(\mathbf{W}^{(0)}, \omega)$. Suppose that $\mathbf{W}^{(0)}, \ldots, \mathbf{W}^{(i)} \in \mathcal{B}(\mathbf{W}^{(0)}, \omega)$. Then by Lemma B.3, for $l \in [L]$ we have $\| \nabla_{\mathbf{W}_l} L_i(\mathbf{W}^{(i)}) \|_F \leqslant \mathcal{O}(\sqrt{m})$. Therefore

$$\left\| \mathbf{W}_l^{(i+1)} - \mathbf{W}_l^{(0)} \right\|_F \leqslant \sum_{j=0}^{i} \left\| \mathbf{W}_l^{(j+1)} - \mathbf{W}_l^{(j)} \right\|_F \leqslant \mathcal{O}(\sqrt{m}\eta n).$$

Plugging in our parameter choice $\eta = \nu\epsilon/(Lm)$, $n = L^2 R^2/(2\nu\epsilon^2)$ for some small enough absolute constant $\nu$ gives

$$\left\| \mathbf{W}_l^{(i+1)} - \mathbf{W}_l^{(0)} \right\|_F \leqslant \mathcal{O}\left( \sqrt{m} \cdot LR^2/(2m\epsilon) \right) \leqslant \omega,$$

where the last inequality holds as long as $m \geqslant C_2 R^{16} L^{56} [\log(m)]^{12} \epsilon^{-14}$ for some large enough constant $C_2$. Therefore by induction we see that $\mathbf{W}^{(0)}, \ldots, \mathbf{W}^{(n-1)} \in \mathcal{B}(\mathbf{W}^{(0)}, \omega)$. As a result, the conditions of Lemmas 4.2 and B.3 are satisfied for $\mathbf{W}^*$ and $\mathbf{W}^{(0)}, \ldots, \mathbf{W}^{(n-1)}$.

In the following, we utilize the results of Lemmas 4.2 and B.3 to prove the bound of cumulative loss. First of all, by Lemma 4.2, we have

$$L_i(\mathbf{W}^{(i-1)}) - L_i(\mathbf{W}^*) \leqslant \langle \nabla_{\mathbf{W}} L_i(\mathbf{W}^{(i-1)}), \mathbf{W}^{(i-1)} - \mathbf{W}^* \rangle + \epsilon$$

$$= \sum_{l=1}^{L} \frac{\langle \mathbf{W}_l^{(i-1)} - \mathbf{W}_l^{(i)}, \mathbf{W}_l^{(i-1)} - \mathbf{W}_l^* \rangle}{\eta} + \epsilon$$

Note that for the matrix inner product we have the equality $2\langle \mathbf{A}, \mathbf{B} \rangle = \|\mathbf{A}\|_F^2 + \|\mathbf{B}\|_F^2 - \|\mathbf{A} - \mathbf{B}\|_F^2$. Applying this equality to the right hand side above gives

$$L_i(\mathbf{W}^{(i-1)}) - L_i(\mathbf{W}^*) \leqslant \sum_{l=1}^{L} \frac{\|\mathbf{W}_l^{(i-1)} - \mathbf{W}_l^{(i)}\|_F^2 + \|\mathbf{W}_l^{(i-1)} - \mathbf{W}_l^*\|_F^2 - \|\mathbf{W}_l^{(i)} - \mathbf{W}_l^*\|_F^2}{2\eta} + \epsilon.$$

By Lemma B.3, for $l \in [L]$ we have $\|\mathbf{W}_l^{(i-1)} - \mathbf{W}_l^{(i)}\|_F \leqslant \eta \|\nabla_{\mathbf{W}_l} L_i(\mathbf{W}^{(i-1)})\|_F \leqslant \mathcal{O}(\eta\sqrt{m})$. Therefore

$$L_i(\mathbf{W}^{(i-1)}) - L_i(\mathbf{W}^*) \leqslant \sum_{l=1}^{L} \frac{\|\mathbf{W}_l^{(i-1)} - \mathbf{W}_l^*\|_F^2 - \|\mathbf{W}_l^{(i)} - \mathbf{W}_l^*\|_F^2}{2\eta} + \mathcal{O}(L\eta m) + \epsilon.$$

Telescoping over $i = 1, \ldots, n$, we obtain

$$\sum_{i=1}^{n} L_i(\mathbf{W}^{(i-1)}) \leqslant \sum_{i=1}^{n} L_i(\mathbf{W}^*) + \sum_{l=1}^{L} \frac{\|\mathbf{W}_l^{(0)} - \mathbf{W}_l^*\|_F^2}{2\eta} + \mathcal{O}(L\eta n m) + n\epsilon$$

$$\leqslant \sum_{i=1}^{n} L_i(\mathbf{W}^*) + \frac{LR^2}{2\eta m} + \mathcal{O}(L\eta n m) + n\epsilon,$$

where in the first inequality we simply remove the term $-\|\mathbf{W}_l^{(n)} - \mathbf{W}_l^*\|_F^2/(2\eta)$ to obtain an upper bound, and the second inequality follows by the assumption that $\mathbf{W}^* \in \mathcal{B}(\mathbf{W}^{(0)}, Rm^{-1/2})$. Plugging in the parameter choice $\eta = \nu\epsilon/(Lm)$, $n = L^2R^2/(2\nu\epsilon^2)$ for some small enough absolute constant $\nu$ gives

$$\sum_{i=1}^{n} L_i(\mathbf{W}^{(i-1)}) \leqslant \sum_{i=1}^{n} L_i(\mathbf{W}^*) + 3n\epsilon,$$

which finishes the proof. $\qquad\square$

### B.4  Proof of Lemma 4.4

Here we prove Lemma 4.4. The proof essentially follows by standard Gaussian tail bound and a bound on the length of last hidden layer output vector.

*Proof of Lemma 4.4.* By Lemma 4.1 in Allen-Zhu et al. [2], with probability at least $1 - \mathcal{O}(nL) \cdot \exp[-\Omega(m/L)] > 1 - \delta/2$ over the randomness of $\mathbf{W}_1^{(0)}, \ldots, \mathbf{W}_{L-1}^{(0)}$, $\|\mathbf{h}_{i,L-1}^{(0)}\|_2 \in [1/2, 3/2]$ for all $i \in [n]$. Condition on $\mathbf{W}_1^{(0)}, \ldots, \mathbf{W}_{L-1}^{(0)}$, $f_{\mathbf{W}^{(0)}}(\boldsymbol{x}_i) = \sqrt{m} \cdot \mathbf{W}_L^{(0)}\mathbf{h}_{i,L-1}$ is a Gaussian random variable with variance $\|\mathbf{h}_{i,L-1}\|_2^2$. Therefore by standard Gaussian tail bound and union bound, with probability at least $1 - \delta$, $|f_{\mathbf{W}^{(0)}}(\boldsymbol{x}_i)| \leqslant \mathcal{O}(\sqrt{\log(n/\delta)})$ for all $i \in [n]$. $\qquad\square$

## C  Proofs of Results in Section A

In this section we provide the proofs of Corollary A.2 and Lemma A.3.

### C.1  Proof of Corollary A.2

The following lemma is a simplified version of Lemma C.2 in Cao and Gu [8]. Since the proof is almost the same as the proof of Lemma C.2 in Cao and Gu [8], except replacing the $\epsilon$-net argument with a simple union bound over $n$ training examples, we omit the proof detail here.

**Lemma C.1.** For any $\delta > 0$, if $m \geqslant K \cdot 4^L L^4 \gamma^{-2} \log(nL/\delta)$ for some large enough absolute constant $K$, then with probability at least $1 - \delta$, there exists $\boldsymbol{\alpha}_{L-1} \in \mathbb{R}^m$ such that $y_i \cdot \langle \boldsymbol{\alpha}_{L-1}, \mathbf{h}_{i,L-1} \rangle \geqslant 2^{-L}\gamma$ for all $i \in [n]$.

*Proof of Corollary A.2.* Set $B = \log\{1/[\exp(n^{-1/2}) - 1]\} = \mathcal{O}(\log(n))$, then for cross-entropy loss we have $\ell(z) \leqslant n^{-1/2}$ for $z \geqslant B$. Moreover, let $B' = \max_{i \in [n]} |f_{\mathbf{W}^{(0)}}(\boldsymbol{x}_i)|$. Then by Lemma 4.4, with probability at least $1 - \delta$, $B' \leqslant \mathcal{O}(\sqrt{\log(n/\delta)})$ for all $i \in [n]$.

By Lemma C.1, with probability at least $1 - \delta$, there exists $\boldsymbol{\alpha}_{L-1} \in S^{m-1}$ such that $y_i \cdot \langle \boldsymbol{\alpha}_{L-1}, \mathbf{h}_{i,L-1} \rangle \geqslant 2^{-L}\gamma$ for all $i \in [n]$. Therefore, setting $R = (B + B') \cdot 2^L\gamma^{-1} = \widetilde{\mathcal{O}}(2^L\gamma^{-1})$, we have

$$\mathbf{W} = (\mathbf{0}, \ldots, \mathbf{0}, Rm^{-1/2} \cdot \boldsymbol{\alpha}_{L-1}^\top) \in \mathcal{B}(\mathbf{0}, Rm^{-1/2}).$$

Moreover, $f^*(\cdot) := f_{\mathbf{W}^{(0)}}(\cdot) + \langle \nabla_{\mathbf{W}} f_{\mathbf{W}^{(0)}}(\cdot), \mathbf{W} \rangle$ satisfies $f^* \in \mathcal{F}(\mathbf{W}^{(0)}, R)$, and

$$
\begin{aligned}
y_i \cdot f^*(\mathbf{x}_i) &= y_i \cdot f_{\mathbf{W}^{(0)}}(\mathbf{x}_i) + y_i \cdot \langle \sqrt{m} \cdot \mathbf{h}_{i,L-1}^\top, Rm^{-1/2} \cdot \boldsymbol{\alpha}_{L-1}^\top \rangle \\
&\geqslant (B + B') \cdot 2^L \gamma^{-1} \cdot 2^{-L} \gamma - B' \\
&\geqslant B.
\end{aligned}
$$

Therefore we have $\ell(y_i \cdot f^*(\mathbf{x}_i)) \leqslant \epsilon, i \in [n]$. Applying Theorem 3.3 gives

$$
\mathbb{E}\big[L_{\mathcal{D}}^{0-1}(\widehat{\mathbf{W}})\big] \leqslant \widetilde{\mathcal{O}}\bigg( \frac{2^L \cdot \gamma^{-1}}{\sqrt{n}} \bigg) + \mathcal{O}\bigg[ \sqrt{\frac{\log(1/\delta)}{n}} \bigg] = \widetilde{\mathcal{O}}\bigg( \frac{2^L \cdot \gamma^{-1}}{\sqrt{n}} \bigg).
$$

This finishes the proof. $\qquad\square$

## C.2 Proof of Lemma A.3

Here we give the proof of Lemma A.3. It is based on a simple construction.

*Proof of Lemma A.3.* For any $f(x) = \mathbf{W}_2 \sigma(\mathbf{W}_1^{(0)} \mathbf{x})$ with $\|\mathbf{W}_2\|_F \leqslant Cm^{-1/2}$, by the assumption that $\|\mathbf{W}_2^{(0)}\|_F \leqslant Km^{-1/2}$ for some $K = \widetilde{\mathcal{O}}(1)$, we have $\mathbf{W}_2' := \mathbf{W}_2 - \mathbf{W}_2^{(0)}$ satisfies $\|\mathbf{W}_2'\|_F \leqslant (C + K) \cdot m^{-1/2}$. Therefore

$$
f(x) = \mathbf{W}_2 \sigma(\mathbf{W}_1^{(0)} \mathbf{x}) = \mathbf{W}_2^{(0)} \sigma(\mathbf{W}_1^{(0)} \mathbf{x}) + \mathbf{W}_2' \sigma(\mathbf{W}_1^{(0)} \mathbf{x}) \subseteq \mathcal{F}.
$$

This finishes the proof. $\qquad\square$

# D Proofs of Lemmas in Section B

In this section we give the proofs of lemma B.1, Lemma B.2 and Lemma B.3 in Section B.

## D.1 Proof of Lemma B.1

*Proof of Lemma B.1.* By Lemma 4.1 in Allen-Zhu et al. [2], with probability at least $1 - \mathcal{O}(nL) \cdot \exp[-\Omega(m/L)]$, $\|\mathbf{h}_{i,l}^{(0)}\|_2 \in [3/4, 5/4]$ for all $i \in [n]$ and $l \in [L-1]$. Moreover, by Lemma 5.2 in Allen-Zhu et al. [2] and the 1-Lipschitz continuity of $\sigma(\cdot)$, with probability at least $1 - \mathcal{O}(nL) \cdot \exp[-\Omega(m\omega^{2/3}L)]$, $\|\mathbf{h}_{i,l} - \mathbf{h}_{i,l}^{(0)}\|_2 \leqslant \mathcal{O}(\omega L^{5/2}\sqrt{\log(m)})$. Therefore by the assumption that $\omega \leqslant \mathcal{O}(L^{-9/2}[\log(m)]^{-3})$, we have $\|\mathbf{h}_{i,l}\|_2 \in [1/2, 3/2]$ for all $i \in [n]$ and $l \in [L-1]$. $\qquad\square$

## D.2 Proof of Lemma B.2

We first introduce the following lemma characterizing the activation changes between networks with two close enough parameter sets $\mathbf{W}$ and $\mathbf{W}'$. This lemma directly follows by Lemma 8.2 in Allen-Zhu et al. [2] and triangle inequality.

**Lemma D.1.** If $\omega \leqslant \mathcal{O}(L^{-9/2}[\log(m)]^{-3/2})$, then with probability at least $1 - \mathcal{O}(nL) \cdot \exp[-\Omega(m\omega^{2/3}L)]$,

$$
\|\mathbf{D}_{i,l} - \mathbf{D}_{i,l}'\|_0 \leqslant \mathcal{O}(L\omega^{2/3}m)
$$

for all $\mathbf{W}, \mathbf{W}' \in \mathcal{B}(\mathbf{W}^{(0)}, \omega), i \in [n]$ and $l \in [L-1]$.

*Proof of Lemma B.2.* We first prove (i) and (iii), and then use (iii) to prove (ii).

By Lemma D.1, with probability at least $1 - \mathcal{O}(nL) \cdot \exp(-\Omega(L\omega^{2/3}m))$, $\|\mathbf{D}_{i,l} - \mathbf{D}_{i,l}^{(0)}\|_0 \leqslant \mathcal{O}(L\omega^{2/3}m)$ for all $i \in [n]$ and $l \in [L-1]$. Therefore we have $\|\mathbf{D}_{i,r} + \mathbf{D}_{i,r}'' - \mathbf{D}_{i,l}^{(0)}\|_0 \leqslant \mathcal{O}(L\omega^{2/3}m)$ for all $i \in [n]$ and $l \in [L-1]$. Therefore by Lemma 5.6 in Allen-Zhu et al. [2], with probability at least $1 - \mathcal{O}(nL^2) \cdot \exp[-\Omega(m\omega^{2/3}L)]$ we have $\big\| \prod_{r=l_1}^{l_2} (\mathbf{D}_{i,r} + \mathbf{D}_{i,r}'') \mathbf{W}_r \big\|_2 \leqslant \mathcal{O}(\sqrt{L})$. This completes the proof of (i) in Lemma B.2.

Similarly, to prove (iii), applying Lemma D.1 to $\mathbf{W}'$ gives that with probability at least $1 - \mathcal{O}(nL) \cdot \exp(-\Omega(L\omega^{2/3}m))$, $\|\mathbf{D}'_{i,l} + \mathbf{D}''_{i,r} - \mathbf{D}^{(0)}_{i,l}\|_0 \leqslant \mathcal{O}(L\omega^{2/3}m)$ for all $i \in [n]$ and $l \in [L-1]$. Now by Lemma 5.7 in Allen-Zhu et al. [2][4] with $s = \mathcal{O}(m\omega^{2/3}L)$ to $\mathbf{W}$ and $\mathbf{W}'$, we have

$$\sqrt{m} \cdot \left\| \mathbf{W}^{(0)}_L \prod_{r=l_1}^{L-1} (\mathbf{D}'_{i,r} + \mathbf{D}''_{i,r})\mathbf{W}'_r - \mathbf{W}^{(0)}_L \prod_{r=l_1}^{L-1} \mathbf{D}^{(0)}_{i,r}\mathbf{W}^{(0)}_r \right\|_2 \leqslant \mathcal{O}\left(\omega^{1/3}L^2\sqrt{m\log(m)}\right), \quad \text{(D.1)}$$

$$\sqrt{m} \cdot \left\| \mathbf{W}^{(0)}_L \prod_{r=l_1}^{L-1} \mathbf{D}_{i,r}\mathbf{W}_r - \mathbf{W}^{(0)}_L \prod_{r=l_1}^{L-1} \mathbf{D}^{(0)}_{i,r}\mathbf{W}^{(0)}_r \right\|_2 \leqslant \mathcal{O}\left(\omega^{1/3}L^2\sqrt{m\log(m)}\right). \quad \text{(D.2)}$$

Moreover, by result (i), we have

$$\left\| (\mathbf{W}'_L - \mathbf{W}^{(0)}_L) \prod_{r=l_1}^{L-1} (\mathbf{D}'_{i,r} + \mathbf{D}''_{i,r})\mathbf{W}'_r \right\|_2 \leqslant \mathcal{O}(\sqrt{L}\omega) \leqslant \mathcal{O}\left(\omega^{1/3}L^2\sqrt{\log(m)}\right), \quad \text{(D.3)}$$

$$\left\| (\mathbf{W}_L - \mathbf{W}^{(0)}_L) \prod_{r=l_1}^{L-1} \mathbf{D}_{i,r}\mathbf{W}_r \right\|_2 \leqslant \mathcal{O}(\sqrt{L}\omega) \leqslant \mathcal{O}\left(\omega^{1/3}L^2\sqrt{\log(m)}\right). \quad \text{(D.4)}$$

Combining equations (D.1), (D.2), (D.3), (D.4) and applying triangle inequality gives the desired final result (iii).

Finally to prove (ii), we write

$$\left\| \mathbf{W}_L \prod_{r=l_1}^{L-1} (\mathbf{D}_{i,r} + \mathbf{D}''_{i,r})\mathbf{W}_r \right\|_2 \leqslant \left\| \mathbf{W}_L \prod_{r=l_1}^{L-1} (\mathbf{D}_{i,r} + \mathbf{D}''_{i,r})\mathbf{W}_r - \mathbf{W}^{(0)}_L \prod_{r=l_1}^{L-1} \mathbf{D}^{(0)}_{i,r}\mathbf{W}^{(0)}_r \right\|_2$$
$$+ \left\| \mathbf{W}^{(0)}_L \prod_{r=l_1}^{L-1} \mathbf{D}^{(0)}_{i,r}\mathbf{W}^{(0)}_r \right\|_2 .$$

Applying (iii) and (b) in Lemma 4.4 in Allen-Zhu et al. [2], with probability at least $1 - \mathcal{O}(nL) \cdot \exp[-\Omega(m/L)]$, we obtain

$$\left\| \mathbf{W}_L \prod_{r=l_1}^{L-1} (\mathbf{D}_{i,r} + \mathbf{D}''_{i,r})\mathbf{W}_r \right\|_2 \leqslant \mathcal{O}\left(\omega^{1/3}L^2\sqrt{\log(m)}\right) + \mathcal{O}(1) = \mathcal{O}(1).$$

This gives (ii). $\qquad\square$

### D.3  Proof of Lemma B.3

*Proof of Lemma B.3.* By Lemma B.1, clearly we have
$$\|\nabla_{\mathbf{W}_l} f_{\mathbf{W}}(\mathbf{x}_i)\|_F = \|\sqrt{m} \cdot \mathbf{h}_{i,L-1}\|_2 \leqslant \mathcal{O}(\sqrt{m})$$
for all $\mathbf{W} \in \mathcal{B}(\mathbf{W}^{(0)}, \omega)$ and $i \in [n]$. For $l \in [L-1]$, by direct calculation we have

$$\|\nabla_{\mathbf{W}_l} f_{\mathbf{W}}(\mathbf{x}_i)\|_F = \sqrt{m} \cdot \left\| \mathbf{h}_{i,l-1}\mathbf{W}_L \left( \prod_{r=l+1}^{L-1} \mathbf{D}_{i,r}\mathbf{W}_r \right) \mathbf{D}_{i,l} \right\|_F$$
$$= \sqrt{m} \cdot \|\mathbf{h}_{i,l-1}\|_2 \cdot \left\| \mathbf{W}_L \left( \prod_{r=l+1}^{L-1} \mathbf{D}_{i,r}\mathbf{W}_r \right) \mathbf{D}_{i,l} \right\|_2 .$$

Therefore by Lemma B.1 and (ii) in Lemma B.2, we have
$$\|\nabla_{\mathbf{W}_l} f_{\mathbf{W}}(\mathbf{x}_i)\|_F \leqslant \mathcal{O}(\sqrt{m}).$$
Finally, for $\|\nabla_{\mathbf{W}_l} L_i(\mathbf{W}^{(i)})\|_F$ we have

$$\|\nabla_{\mathbf{W}_l} L_i(\mathbf{W}^{(i)})\|_F \leqslant \left| \ell'[y_i \cdot f_{\mathbf{W}^{(i)}}(\mathbf{x}_i)] \cdot y_i \right| \cdot \left\| \nabla_{\mathbf{W}_l} f_{\mathbf{W}^{(i)}}(\mathbf{x}_i) \right\|_F \leqslant \sqrt{m}.$$

This completes the proof. $\qquad\square$

# E  Experimental Results

In this section we provide numerical calculations of the generalization bounds given by Theorem 3.3 and Corollary 3.10 on the MNIST dataset [22]. The main goal of these calculations is to demonstrate that the bounds given in our results are informative and can provide practical insight.

We have done experiments of a five-layer fully connected NN on MNIST dataset (3 versus 8), and calculated the first terms in the bounds given by Theorem 3.3 and Corollary 3.10.

- In Figure 1(a), we plot the first term in the bound of Theorem 3.3 with different values of $R$ and $m$, where the infimum of loss function values is approximated by solving the constrained convex optimization problem $\inf_{f \in \mathcal{F}(\mathbf{W}^{(0)}, R)}\{(4/n) \cdot \sum_{i=1}^{n} \ell[y_i \cdot f(\mathbf{x}_i)]\}$ with projected stochastic gradient descent.

- To demonstrate the scaling of the bound in Corollary 3.10, we calculate the value of $\sqrt{\mathbf{y}^\top (\mathbf{\Theta}^{(L)})^{-1} \mathbf{y}/n}$, where $\mathbf{y}$ is the true label vector with random flips. We plot $\sqrt{\mathbf{y}^\top (\mathbf{\Theta}^{(L)})^{-1} \mathbf{y}/n}$ in Figure 1(b) by varying the level of label noise, i.e., ratio of the labels that are flipped. Note that to simplify calculation, we do not consider the $\widetilde{\mathbf{y}}$ introduced in Corollary 3.10. Clearly, our calculation here gives an upper bound of the generalization bound in Corollary 3.10.

Figure 1: (a) Evaluation of the first term in the bound of Theorem 3.3 for different values of $R$ and $m$. (b) Evaluation of the first term of the bound in Corollary 3.10 with different ratio of label flip.

We can see that our bounds in both Theorem 3.3 and Corollary 3.10 give small and meaningful values. Moreover, these experimental results also back up our theoretical analysis. In Figure 1(a), the curves corresponding to different $m$'s also validate our theoretical result that the wider the network is, the shorter SGD needs to travel to fit the training data. In addition, the larger the size of reference function class (i.e., $R$), the smaller $\inf_{f \in \mathcal{F}(\mathbf{W}^{(0)}, R)}\{(4/n) \cdot \sum_{i=1}^{n} \ell[y_i \cdot f(\mathbf{x}_i)]\}$ will be. In Figure 1(b), we can see that the noisier the labels, the larger the term $\sqrt{\mathbf{y}^\top (\mathbf{\Theta}^{(L)})^{-1} \mathbf{y}/n}$ is. When most of the labels are true labels, our bound can predict good test error; when the labels are purely random (i.e., ratio of label flip $= 0.5$), the bound on the test error can be larger than one. To sum up, these numerical results demonstrate the practical values of our generalization bounds, and suggest that our bounds can provide good measurements of the data classifiability.

## Footnotes

[4]Note that $\sqrt{m} \cdot \mathbf{W}^{(0)}_L$ is a random vector following the Gaussian distribution $N(\mathbf{0}, \mathbf{I})$, which matches the distribution of the last layer parameters in Allen-Zhu et al. [2] for the binary classification case, where the output dimension of the network is 1.