[Reviews · NeurIPS 2019]

Reviewer 1



This paper studies the task of training over-parameterized deep ReLU networks via SGD, and proves a generalization bound which is more general or sharper than several recent ones. The generalization bound has the additional nice property which can distinguish between learning from noisy labels and true ones. The analysis is intuitive and seems to take a slightly different route from most previous works I am aware of, which may benefit the future work on this topic. Therefore, I feel that the paper has enough originality and significance. The results appear technically sound to me, as the proofs all look reasonable, although I did not check them very carefully. The paper is well written and easy to follow in general. * After author response: I have read the response, and I am satisfied with it. I still keep my score as “Marginally above the acceptance threshold” and vote for accepting the paper.

Reviewer 2



Originality: To the best of my knowledge, the results are novel and provide important extensions/improvements over the previous art. Quality: I did a high level check of the proofs and it seems sound to me. Clarity: the paper is a joy to read. The problem definition, assumptions, the algorithm, and statement of results are very well presented. Significance: the results provide several extensions and improvements over the previous work, including training deeper models, training all layers, training with SGD (rather than GD), and smaller required overparameterization. More importantly, the proofs are simpler compared to the related work. My only concern are the followings. 1) the width requirement is still very stringent, and not realistic; and 2) It is not clear how to compare such a bound with other results. In particular, it would have been nice to see how small the NTRK error (the first term in the right hand side of Thm 3.3.) can get in practice for real models. ******************************************************************** *******************after author feedback********************** ******************************************************************** I have read the authors feedback and am happy to increase my rating from 6 to 8. I have also increased my confidence score from 3 to 4 after authors clarified some part of the proof. Overall, I think this paper is well qualified to be published in NeurIPS.

Reviewer 3



The authors describe the proof for expected 0-1 error of deep ReLU networks trained with SGD using random initialization (from He. et. al.), that results in algorithm dependent generalization error bound that is independent of network width if the data can be classified by the purposed NTRF features with small enough error. The result is generalizable (two layer and larger function class) and when reduced to two layer setting is sharper. A more general and tighter bound is also derived, similar to Arora [3] to show similarity to neural tangent kernel from Jacot [17]. The purposed NTRF is shown to be richer (contains gradient from first layer) and generalization bound sharper ( compared to 31 and 13). He initialization enables removing exponential dependence on depth in kernel matrix, where the kernel matrix takes all layers into consideration (as opposed to last hidden layer in [10]). The results are also compared and shown as generalization of [3] with sharper bounds.

[Author Response · NeurIPS 2019]

**Response to Reviewer #1:**

**Q1.** "The result establishes a connection to some kernel method in previous work. Significance: low."

**A1.** We clarify that our result in Section 3.2 is not a re-derivation of existing result. To our knowledge, existing results on NTK all focus on square loss, while the connection between NTK and NNs trained by minimizing cross-entropy loss is previously unknown. Therefore our results on the connection to NTK is still new and significant.

**Q2.** "The generalization bound is only shown for the network at a randomly chosen step... any of the final step"

**A2.** Our generalization bound at a randomly chosen step matches the standard results for stochastic optimization. Our result also directly implies bounds on the 'best iterate'. To the best of our knowledge, previous works on generalization bounds of SGD-trained NNs, including Daniely [9], Allen-Zhu et al. [1] and Yehudai and Shamir [31], are all essentially of the same type, i.e., either on a randomly chosen step, or on the 'best iterate'. We noticed that very recently [*] established the last iterate bound of SGD for convex optimization with decreasing step sizes. However, it is still not clear whether the last iterate guarantee can be proved for SGD-trained NNs, which is essentially a nonconvex (almost convex) optimization problem. We will study it in our future work.

*[*] Jain, P., Nagaraj, D. and Netrapalli, P., Making the last iterate of SGD information theoretically optimal, COLT'19.*

**Q3.** "... how the over-parameterization requirement of this paper compares to those in related works."

**A3.** To the best of our knowledge, the over-parameterization condition $m = \Omega(n^7)$ in this paper is the mildest compared with existing results for *deep* ReLU networks. (Note that many results' over-parameterization conditions are dependent on the smallest eigenvalue of a kernel matrix, which hides dependency in $n$.) We will add this remark in our revision.

**Response to Reviewer #2:**

**Q1.** "... width requirement is still very stringent"

**A1.** We agree that the condition on the number of hidden nodes per layer is still large, compared with the number of hidden nodes used in practice. Nevertheless, to the best of our knowledge, our over-parameterization condition is already the mildest among existing results for ReLU networks. Moreover, for smooth activation functions, we can further improve the condition to be $m = \Omega(n^2)$.

**Q2.** "... proof of Lemma 4.2. page 13, line 464... bound."

**A2.** We clarify that the proof is correct. By chain rule, we have $\sum_{l=1}^{L}\langle\nabla_{\mathbf{W}_l}L_i(\mathbf{W}),\mathbf{W}'_l-\mathbf{W}_l\rangle = \ell'[y_if_{\mathbf{W}}(\mathbf{x}_i)]\cdot y_i\cdot\langle\nabla f_{\mathbf{W}}(\mathbf{x}_i),\mathbf{W}'-\mathbf{W}\rangle$. Therefore by triangle inequality, the RHS of the inequality above line 464 has the lower bound

Figure 1: (a) Evaluation of the first term in the bound of Theorem 3.3 for different values of $R$ and $m$. (b) Evaluation of the first term of the bound in Corollary 3.10 with different ratio of label flip.

$$\ell'[y_if_{\mathbf{W}}(\mathbf{x}_i)]\cdot y_i[f_{\mathbf{W}'}(\mathbf{x}_i) - f_{\mathbf{W}}(\mathbf{x}_i)] \geqslant \ell'[y_if_{\mathbf{W}}(\mathbf{x}_i)]\cdot y_i\langle\nabla f_{\mathbf{W}}(\mathbf{x}_i),\mathbf{W}'-\mathbf{W}\rangle - I = \sum_{l=1}^{L}\langle\nabla_{\mathbf{W}_l}L_i(\mathbf{W}),\mathbf{W}'_l-\mathbf{W}_l\rangle - I,$$

where $I = \left|\ell'[y_if_{\mathbf{W}}(\mathbf{x}_i)]\cdot y_i\cdot\left[f_{\mathbf{W}'}(\mathbf{x}_i) - f_{\mathbf{W}}(\mathbf{x}_i) - \langle\nabla f_{\mathbf{W}}(\mathbf{x}_i),\mathbf{W}'-\mathbf{W}\rangle\right]\right|$. The first inequality below line 464 then follows by upper-bounding $I$ with Lemma 4.1 and the fact that $|\ell'[y_if_{\mathbf{W}}(\mathbf{x}_i)]\cdot y_i| \leqslant 1$.

**Q3.** "... small scale experiments evaluating the first term in the RHS of Thm 3.3. and Corr 3.10..."

**A3.** Following your suggestion, we have done experiments of a five-layer fully connected NN on MNIST dataset (3 versus 8), and calculated the first terms in the bounds given by Theorem 3.3 and Corollary 3.10. In particular, we plot the first term in the bound of Theorem 3.3 in Figure 1(a), by varying the values of $R$ and $m$. We can see that our bound gives small and meaningful values. The curves corresponding to different $m$'s also validates our theoretical result that the wider the network is, the shorter SGD needs to travel to fit the training data. In addition, the larger the size of reference function class (i.e., $R$), the smaller this term will be. In addition, we plot the first term in the bound of Corollary 3.10 in Figure 1(b) by varying the level of label noise, i.e., ratio of the labels that are flipped. We can see that the noisier the labels, the larger this term is. When most of the labels are true labels, our bound can predict good test error; when the labels are purely random (i.e., ratio of label flip = 0.5), the bound on the test error can be larger than one. These plots demonstrate the practical values of our generalization bounds, and suggest that our bound can provide good measurements of the data classifiability. We will add these experimental results in the camera ready.

**Q4.** "Suggestion: the connection to NTK is rather straightforward... in the first page? "

**A4.** Thanks for the suggestion. At high level, if data are generated by $y = f^*(\mathbf{x})$ for $f^*(\cdot)$ with bounded norm in the NTK-induced RKHS space, SGD-trained NNs generalizes well. We will add more discussions and examples.

**Response to Reviewer #3:**

**Q1.** "Some treatment of the neural tangent random feature limitations... random initialization. (lines 159-161)"

**A1.** We clarify that the infimum on the right-hand of theorem 3.3 is a convex optimization problem that only depends on training data and can be easily solved. Therefore the bound can be easily calculated in practice. We will add discussion and examples on the target function $y = f^*(\mathbf{x})$ that can be learned by NNs trained with SGD in the revision.

**Q2.** "There are several statements made without proof... inclusion of experimental results would also help. "

**A2.** We will provide more details on extensions to networks with different layer widths and different loss functions. We will also add experimental results in the revision. Please see Figure 1 above and **A3** to **Reviewer #2**.

[Meta-Review · NeurIPS 2019]

This paper provides a generalization bound for training over-parameterized deep neural networks with ReLU activation and cross-entropy loss using SGD. Initially the paper received mixed reviews, with two positive and one negative reviews. On the one hand, the analysis is found to be intuitive, general, and potentially influential, the generalization bound is found to be more general and sharper than many existing generalization error bounds for over-parameterized neural networks, and the paper to be very well written. On the other, hand the width requirement is found to be too strict. The rebuttal addressed the issues raised by the reviewers, one rating was increased from 6 to 8, and the negative review updated the score to 6. Upon discussion, the reviewers agreed that the paper should be accepted. Overall, this is an excellent paper, that addresses a very important and difficult problem in the theory of deep learning.